# Fluorine Controls Mineral Assemblages of Alkaline Metasomatites

**Julia A. Mikhailova \*, Yakov A. Pakhomovsky, Natalia G. Konopleva, Andrey O. Kalashnikov and Victor N. Yakovenchuk**

Geological Institute of the Kola Science Centre, Russian Academy of Sciences, 14 Fersman Street, 184209 Apatity, Russia

\* Correspondence: mikhailova@geoksc.apatity.ru; Tel.: +7-81555-79333

**Abstract:** In the Khibiny and Lovozero alkaline massifs, there are numerous xenoliths of the so-called 'aluminous hornfelses' composed of uncommon mineral associations, which, firstly, are ultra-aluminous, and secondly, are highly reduced. (K,Na)-feldspar, albite, hercynite, fayalite, minerals of the phlogopite-annite and cordierite-sekaninaite series, corundum, quartz, muscovite, sillimanite, and andalusite are rock-forming minerals. Fluorite, fluorapatite, ilmenite, pyrrhotite, ulvöspinel, troilite, and native iron are characteristic accessory minerals. The protolith of these rocks is unknown. We studied in detail the petrography, mineralogy, and chemical composition of these rocks and believe that hornfelses were formed as a result of the metasomatic influence of foidolites. The main reason for the formation of an unusual aluminous association is the high mobility of aluminum promoted by the formation of fluid expelled from foidolites of the Na-Al-OH-F complexes. Thus, it is fluorine that controls the mobility of aluminum in the fluid and, consequently, the mineral associations of alkaline metasomatites. The gain of alkalis and aluminum to rocks of protolith was the reason for the intense crystallization of (K,Na)-feldspar. As a result, a $SiO_2$ deficiency was formed, and Si-poor, Al-rich silicates and/or oxides crystallized.

**Keywords:** Lovozero massif; Khibiny massif, aluminous hornfelses, alkaline metasomatism; foidolites

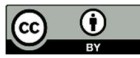

## 1. Introduction

Intrusions of both carbonatite and alkaline rocks are typically found in close spatial relationships with aureoles of high-temperature, metasomatically-altered country rocks, termed fenites. The process of fenitization is generally viewed to result from multiple pulses of alkali-rich fluid expelled from a cooling crystallizing intrusion [1]. Fenites are characterized by the addition of alkalis, volatiles, albitization, nephelinization, removal of silica, and the formation of alkali pyroxenes and amphiboles [2–5]. Such rocks were first described by Brögger in his pioneering work on the rocks of the Fen complex in southern Norway [6]. He defined fenites as a suite of rocks of originally granitic composition that had been metasomatically altered towards an alkali-syenitic composition by solutions sourced from ijolite–melteigite magma within the complex. The term 'fenite' has since taken on a more general meaning and encompasses a wide spectrum of alkaline alteration products developed near silica-undersaturated, alkaline intrusions. Mineral assemblages of fenites are highly variable and dependent on a number of parameters such as protolith mineralogy, permeability and structure, compositions of alkaline melt and fluid, temperature, and pressure [7–10].

A remarkable example of the dependence of the intensity of fenitization and mineral associations of fenites on the above-mentioned parameters is the Khibiny alkaline massif (Kola Peninsula, Russia). In the north, northeast, and southeast, the massif is in contact

with Archean gneisses and migmatite. In the west, southwest, and south, Proterozoic greenstone belt rocks, namely greenschist, meta-diabase, meta-gabbro, tuffaceous schist, pillow lava, and granophyre, are in contact with the massif [11,12]. Correspondingly, the mineral associations formed as a result of the alteration of country rocks under the influence of alkaline intrusion are very different. During the fenitization of gneisses, 'classical' fenites were formed, mainly consisting of microcline, alkali pyroxenes, and amphiboles [13,14], while greenstone rocks were hornfelsed without a significant change in chemical composition [14]. Numerous xenoliths of both metasomatized Archean gneisses and greenstone rocks were found among alkaline rocks near the massif's contacts.

However, xenoliths are located not only near contacts with country rocks. In the internal parts of the Khibiny massif, there are a large number of xenoliths of the so-called 'aluminous hornfels', composed of uncommon mineral associations, including, in addition to alkali feldspar and albite, hercynite, fayalite, minerals of the phlogopite-annite and cordierite-sekaninaite series, corundum, quartz, muscovite, sillimanite, and andalusite [12]. Fluorite, fluorapatite, ilmenite, pyrrhotite, sphalerite, titaniferous magnetite, ulvöspinel, troilite, and native iron are characteristic accessory minerals. Such mineralogy is not observed either in fenitized gneisses or in hornfelsed greenstone rocks [14]. Similar hercynite- and sekaninaite-bearing aluminous hornfels are also found within the Lovozero alkaline massif, located near the Khibiny massif and surrounded by Archean gneisses. What rock was the protolith of aluminous hornfels, and under what conditions were mineral associations of these rocks formed? These issues are debatable in the present day, despite the long history of studying the Khibiny and Lovozero alkaline massifs [11,12,15–18].

In this article, we present the results of a study of the petrography, mineralogy, and chemical composition of aluminous hornfels from xenoliths in the Khibiny and Lovozero massifs, and form conclusions about the reasons and conditions for the formation of these unusual rocks.

## 2. Geological Background and Previous Research

The Khibiny and Lovozero alkaline massifs are located at the southwest of the Kola Peninsula and occupy areas of 1327 and 650 km$^2$, respectively. The Lovozero massif intrudes the Archean gneisses of the Kola–Norvegian block, while the Khibiny massif is located at the contact of the Archean rocks and the Proterozoic Pechenga–Imandra–Varzuga greenstone belt (Figure 1a). The ages of the major rock types of the Khibiny and Lovozero massifs were determined, respectively, as 360–380 Ma [19] and 360–370 Ma [20–22].

The Khibiny and Lovozero massifs are composed mainly of nepheline syenites and foidolites [11,16,23,24]. Some varieties of nepheline syenites in these massifs have historically accepted local names [25] that are important for describing the geology of the massifs, and we will use them in the text below. These rock names are as follows:

- *foyaite* is a massive, less often weakly trachytoid, leucocratic nepheline syenite;
- *rischorrite* is a leucocratic nepheline syenite in which the nepheline crystals are poikilitically enclosed in microcline perthite;
- *lyavochorrite* is a leucocratic nepheline syenite in which only part of the feldspar crystals is poikilitic;
- *lujavrite* is a trachytoid (i.e., with subparallel feldspar laths) meso- or melanocratic nepheline syenite.

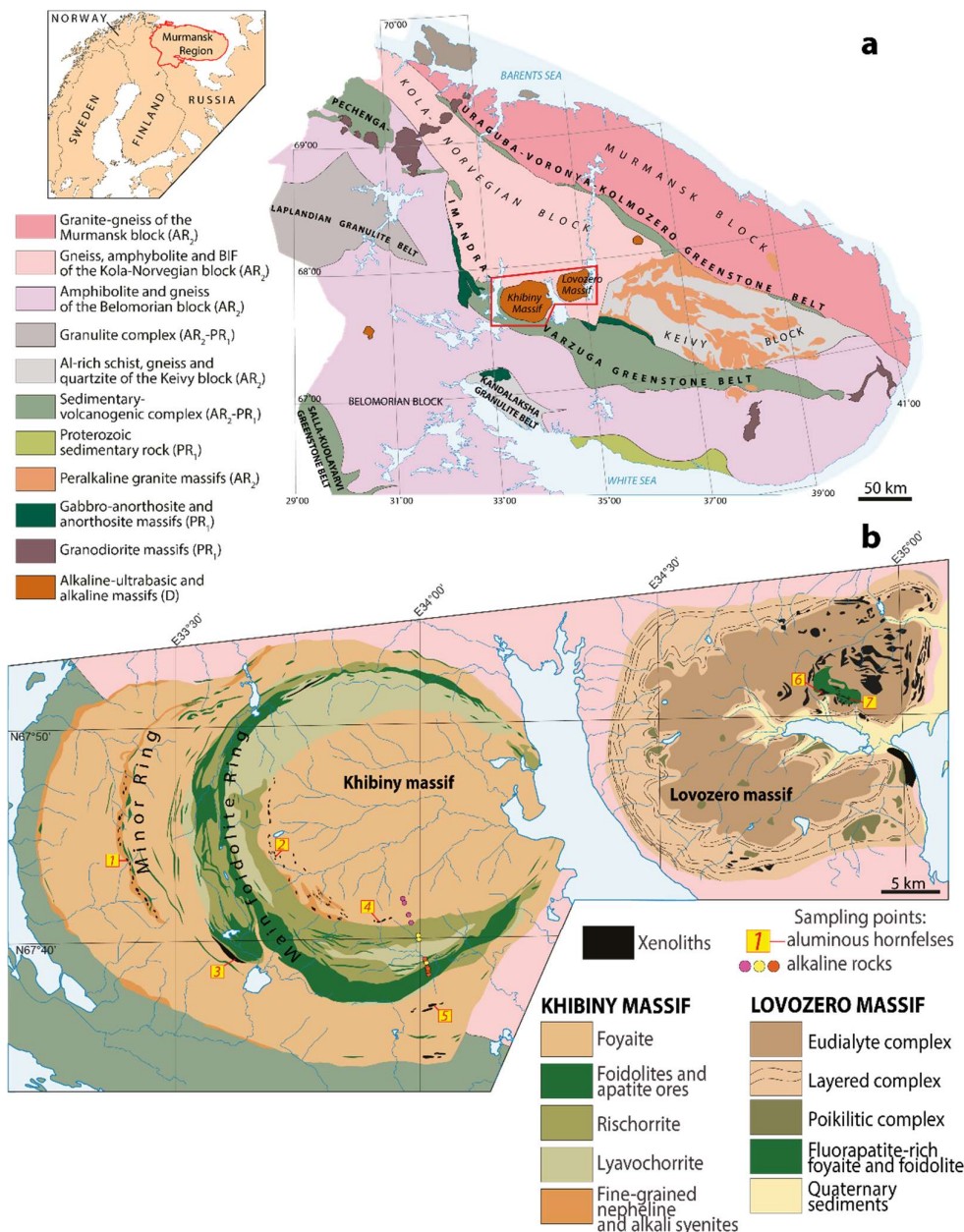

**Figure 1.** (**a**) Simplified geological map of the Murmansk Region [26]; (**b**) geological schemes of the Khibiny and Lovozero massifs [27,28].

The Khibiny massif has a concentrically zoned structure (Figure 1b). In plain view, the massif is elliptical (45 km × 35 km) and vertically it is cone-like, with its apex pointing downward [29]. The massif consists dominantly of foyaite (about 70% of the outcrop area) and foidolites (mainly ijolite and urtite, 8% of the outcrop area) that intruded into the foyaite along the two cone-like faults: the Main Ring and Minor Ring faults [15,24,27]. Poikilitic nepheline syenites, namely rischorrite (10% of the outcrop area) and lyavochorrite (9% of the outcrop area) occur between the rocks of the Main Ring and the foyaite. The foidolites of the Main Ring accommodate all the apatite deposits and occurrences. The apatite-nepheline and titanite-apatite-nepheline ores form stockworks in the apical parts of the foidolite intrusions.

The Lovozero massif is a layered laccolith [17,28]. According to geophysical studies [29], alkaline rocks are traced to a depth of 7 km, and the lower limit of their distribution

is not detected. The laccolith has a size of 20 km × 30 km at the top, and about 12 km × 16 km at a 5 km depth [17]. In the upper part, the intrusion contacts with host rocks are almost vertical. The Lovozero massif consists of three main units (Figure 1b): Layered, Eudialyte, and Poikilitic complexes. The Layered complex comprises 77% of the massif volume [17], has a thickness of more than 1700 m, and consists of numerous layers (or rhythms). The idealized rhythm is a sequence of rocks (from top to bottom): lujavrite–foyaite–urtite [30]. The transition between rocks within the rhythm is gradual, and contacts between the rhythms are sharp. The Eudialyte complex is located in the upper part of the Lovozero massif (Figure 1b) and occupies 18% of its volume [17]. The thickness of this complex is from 100 m (in the east) to 800 m (in the northwest). The main rock type of the Eudialyte complex is lujavrite enriched in eudialyte-group minerals—the so-called eudialyte lujavrite. Among eudialyte lujavrite, lenses and layers of foyaite, as well as fine-grained and porphyritic nepheline syenites, are irregularly located. The poikilitic complex (5% massif's volume) consists of feldspathoid (nepheline, sodalite, or vishnevite) syenites. These rocks form lenses, or irregularly shaped bodies, which are located in both the Layered and Eudialyte complexes.

There are many dissimilar xenoliths among the alkaline rocks of the Khibiny and Lovozero massifs. All xenoliths can be divided into two main groups: (1) endocontact xenoliths and (2) xenoliths in the internal parts of the massifs. Endocontact xenoliths are fenitized Archean gneiss (both in the Khibiny and Lovozero massifs) and greenstone rocks (only in the Khibiny massif). The largest amount of such xenoliths was found at a distance of several tens of meters from the contact, but some were found inside the massif at a distance of 200–300 m from the contact. The petrography, mineralogy, chemical composition, and metasomatic alteration of the rocks of the endocontact xenoliths were studied in detail in many works [12,14,31].

Numerous xenoliths are also found in the internal parts of the massifs, far from contacts with country rocks. Such xenoliths are located in accordance with the geological structure of each of the plutons. So, in the Khibiny (concentrically zoned) massif, there are two semicircular zones with the largest number of xenoliths. The first zone is located near the Main foidolite Ring between foyaite and rischorrite/lyavochorrite, and the second zone is placed within the Minor Ring (Figure 1b). In the Lovozero (layered) massif, the xenoliths are subhorizontal sheets or lens-like bodies among the alkaline rocks of the Eudialyte and Layered complexes.

As a rule, individual xenoliths are not very large—about 1–10 m across (Figure 2a,b); they almost never occur separately but form clusters. Therefore, the xenolith-bearing areas in Khibiny and Lovozero can be compared with a giant breccia, where the cement is alkaline rocks. Figure 1b shows only the largest xenoliths, but such bodies are always accompanied by many smaller neighbors. An example of a large (100 m × 50 m) xenolith is that located on Kaskasnyunchorr Mt. in the Khibiny massif (Figure 2c).

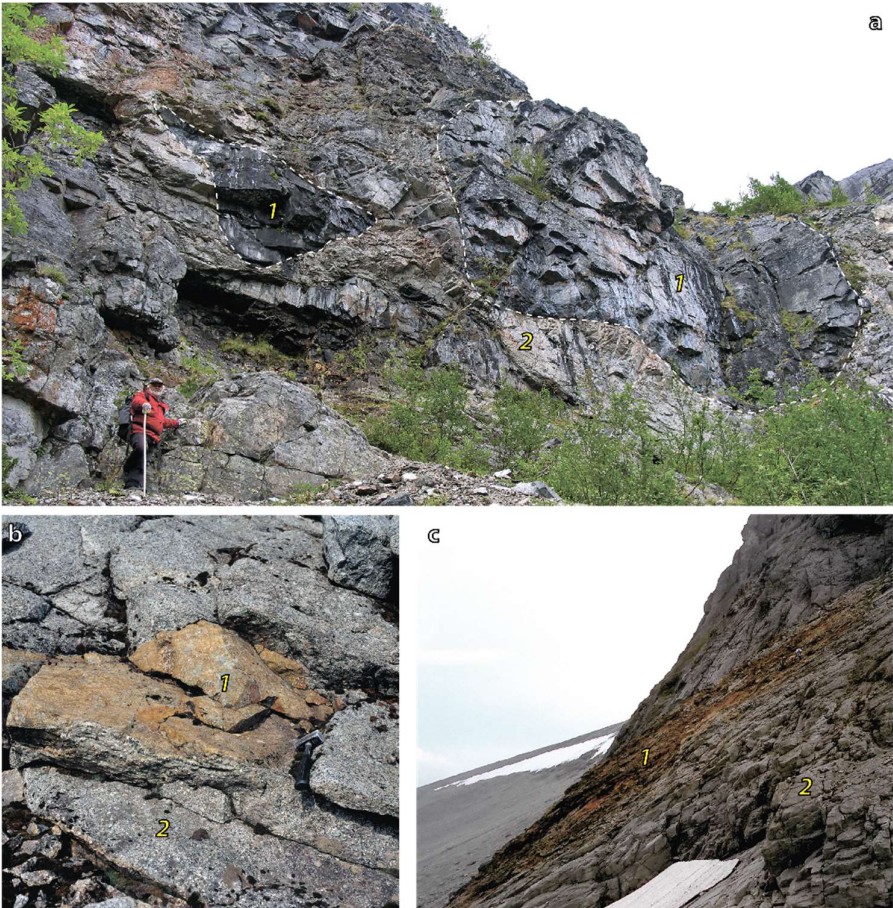

**Figure 2.** Xenoliths of the aluminous hornfelses (1) among alkaline rocks (2). (**a**) on the Kuivchorr (Lovozero massif); (**b**) on the Yudychvumchorr Mt. (Khibiny massif); the size of this xenolith is 1.5 m × 0.6 m; (**c**) on the Kaskasnyunchorr Mt. (Khibiny massif); the size of this xenolith is 100 m × 50 m. Photos by Gregory Ivanuyk.

The petrography of the xenoliths located in the internal parts of the massifs is very diverse. The Lovozero massif contains xenoliths of both aluminous hornfelses and volcaniclastic rocks. The latter are irregularly interbedded olivine basalt, basalt tuff, tuffite, and quartzite [17,32,33]. The rock-forming minerals of basalt and basalt tuff are diopside–augite, plagioclase (oligoclase–andesine), forsterite, and phlogopite. Quartzite and sandstone consist mainly of rounded or angular quartz and microcline grains. Tuffite with variable proportions of pyroclastic and terrigenous materials is an intermediate rock between tuff on the one hand and quartzite and sandstone on the other hand [32]. The age of these lithologies was determined from paleontological data [33,34] and the K-Ar method [35] as Devonian (378 Ma), and therefore it is assumed that the formation of these rocks immediately preceded the emplacement of the Lovozero pluton [17], and now Devonian volcaniclastic rocks are the roof remnants. The Khibiny massif contains xenoliths of aluminous hornfelses, Devonian volcaniclastic rocks, and Proterozoic greenstone rocks [12].

Based on previous studies [11,12,32,36,37], it can be argued that xenoliths of aluminous hornfelses are spatially associated with foidolites, while xenoliths of volcaniclastic and greenstone rocks are irregularly distributed. This is especially clearly manifested in the Lovozero massif [27]. Here, among the rocks of the Eudialyte complex, which occupy most of the area of the massif, xenoliths are very widespread, while foidolites, on the contrary, are rare. However, only xenoliths that are located close to

foidolites consist of aluminous hornfelses, and xenoliths placed at a relatively long distance from foidolites are composed of slightly fenitized volcaniclastic rocks. Figure 3a shows the geological scheme of a part of the Eudialyte complex (Kuivchorr Mt.) where the xenoliths are spatially associated with foidolites. This is the only locality in the Lovozero massif where xenoliths of aluminous hornfelses have been found. In the Khibiny massif, aluminous hornfelses frame the Main foidolite Ring and are located at the contact of rischorrite and foyaite. Here, xenoliths form a crescent zone with a length of about 20 km [36]. In addition, the majority of the pegmatites and hydrothermal veins are located in the same zone (Figure 3b). Additionally, aluminous hornfelses were found in close spatial association with foidolites within the Minor Ring. At a relatively long distance from the Main and Minor foidolite Rings, for example, in the south of the massif, the xenoliths are composed of Devonian volcaniclastic rocks.

According to Korchak and co-authors [32], the aluminous hornfelses in both the Khibiny and Lovozero massifs were formed as a result of the high-temperature fenitization of Devonian volcaniclastic rocks under the influence of alkaline melts. However, there are no specific temperature estimates in this work, and there are no answers to the questions of (1) why only some xenoliths were transformed into aluminous hornfelses, and (2) what conditions determined the formation of highly reduced mineral associations of these hornfelses. According to Shlykova [12], the xenoliths inside the Khibiny massif are the remains of country rocks, i.e., Archean gneiss and Proterozoic greenstone belt rocks, while the formation of aluminous hornfelses is associated with the local enrichment of the protolith with aluminum and the high-temperature impact of crystallizing intrusion. Yakovleva and co-authors [18,38] state that the protolith of the aluminous hornfelses was Archean aluminous schists, similar to the kyanite schists of the Keivy block (see Figure 1a).

The most important problem in attempts to reconstruct the protolith of aluminous hornfelses is the absence of any metasomatic zonation within the xenoliths of the aluminous hornfelses. Indeed, as shown below, xenoliths are very heterogeneous in mineral and modal compositions, but this heterogeneity is chaotic and is not related to the distance from the contact of the xenolith with the host alkaline rocks. Furthermore, alkaline rocks at contact with xenoliths of aluminous hornfelses vary widely in texture and modal composition. The xenoliths are surrounded by aureoles of fine-grained and uneven-grained (large nepheline crystals in a fine-grained mass) foyaites, as well as alkaline syenite. Additionally, alkaline rocks near contact with xenoliths are enriched in titanite, fluorapatite, and pyrrhotite.

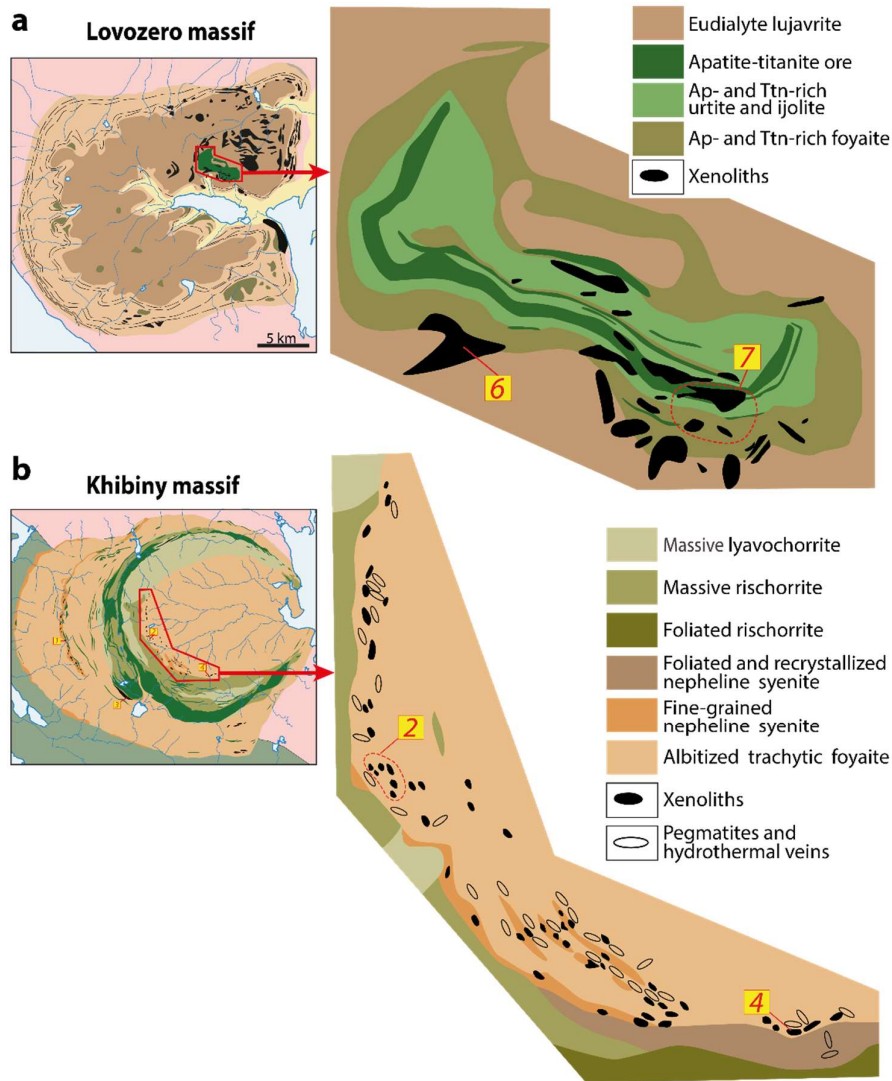

**Figure 3.** Schemes of the location of xenoliths of aluminous hornfelses in the Lovozero and Khibiny massifs. (**a**) part of the Eudialyte complex of the Lovozero massif (outlined in red), where xenoliths of the aluminous hornfelses are located [27]; (**b**) xenoliths of the aluminous hornfelses in the central part of the Khibiny massif [39]. 2, 4, 6, 7 are sampling points (see Figure 1b).

## 3. Materials and Methods

For this study, 88 samples of the aluminous hornfelses were sampled from xenoliths in the Khibiny and Lovozero massifs. The sampling points are shown in Figure 1b. The list of samples and a short description of the studied xenoliths are presented in Table 1.

The 88 thin polished sections were analyzed at the Geological Institute of the Kola Science Center of the Russian Academy of Sciences (GI KSC RAS, Apatity, Russia) using a scanning electron microscope LEO-1450 (Carl Zeiss Microscopy, Oberkochen, Germany) with the energy-dispersive system Quantax 200 and Aztec Ultimmax 100 (Oxford Instruments, UK) to obtain BSE (back-scattered electron) images and pre-analyze all detected minerals. The chemical composition of minerals was analyzed with the Cameca MS-46 electron microprobe (Cameca, Gennevilliers, France) operating in the WDS-mode at 22 kV with a beam diameter of 10 μm, beam current of 20–40 nA, and counting times of 10 s (for a peak) and 10 s (for background before and after the peak), with 5–10 counts for every element in each point. The following standards were used: lorenzenite (Na, Ti), pyrope (Al), wollastonite (Si, Ca), fluorapatite (P), $F_{10}S_{11}$ (Fe, S), atacamite (Cl), wadeite

(K), metallic V, $MnCO_3$ (Mn), hematite (Fe), celestine (Sr), $ZrSiO_4$ (Zr), metallic Nb, baryte (Ba), $LaCeS_2$ (La, Ce), $LiPr(WO_4)_2$ (Pr), $LiNd(MoO_4)_2$ (Nd), $LiSm(MoO_4)_2$ (Sm), metallic Hf and Ta, thorite (Th), and metallic U. The analytical precision (reproducibility) of mineral analyses was 0.2–0.05 wt. % (2 standard deviations) for the major element and approximately 0.01 wt. % for impurities. The systematic errors were within the random errors.

**Table 1.** List of samples.

| Massif | Sampling Point | Samples | Short Geological Description |
|---|---|---|---|
| Khibiny | 1 | KH-61-2, KH-61-3, KH-61-4, KH-61-6, KH-61-9, KH-61-11, KH-61-13, KH-61-15, KH-31-1, KH-31-2, KH-32-2, KH-32-3, KH-33-1 | Numerous small, rounded xenoliths located at the contact of foyaite and foidolites. The sizes of individual xenoliths are from 10 cm × 20 cm to 1 m × 3 m. |
| Khibiny | 2 | KCH-05-11, KCH-05-13, KCH-05-15, KCH-05-21, KCH-05-24, KCH-05-31, KC-4, M-01-1-10, M-01-2-12, M-01-2-15, M-01-26, M-01-27, M-01-2-8, M-01-2-9, M-01-3, M-01-2, M-01-2-1, M-01-2-4 | Numerous small xenoliths located near the contact of foyaite and rischorrite. The size of individual xenoliths is on average 2 m × 3 m. |
| Khibiny | 3 | KH-16/86, KH-17/86, KH-18/86 | Samples from a borehole. Very large xenolith (600 m × 3000 m × 6000 m) located at the contact of foyaite and rischorrite [12,16]. |
| Khibiny | 4 | from E-97-1 to E-97-45 | Large (120 m × 40 m) xenolith located near the contact of foyaite and rischorrite. |
| Khibiny | 5 | KH-110, KH-111 | Large (100 m × 40 m) xenolith in foyaite. |
| Lovozero | 6, 7 | LV-117, LV-119, LV-119A, LV-120, LV-121, LV-121A, LV-122, LV-132, LV-149/2, LV-150/2, LV-160/1, LV-160/4, LV-01-45 | Several xenoliths located among fluorapatite-enriched foyaite and foidolites. The sizes of individual xenoliths are from 0.5 m × 1 m to 2 m × 5 m. |

Diagnostics of the $Al_2(SiO_4)O$ polymorphs were carried out using X-ray diffraction (XRD). XRD measurements were taken with a DRON-2 diffractometer (Burevestnik, Saint-Peterburg, Russia) at the GI KSC RAS. The operating parameters were as follows: $CuK\alpha$ radiation, 20 mA, 30 kV. XRD data were identified using the RRUFF Project database [40].

Major elements in rocks (45 samples) were determined by wet chemical analysis at the GI KSC RAS. The accuracy limits for $SiO_2$, $TiO_2$, $ZrO_2$, $Fe_2O_3$, $Al_2O_3$, CaO, SrO, MgO, MnO, $Na_2O$, $K_2O$, $P_2O_5$, $REE_2O_3$, $S_{tot}$, F, Cl, $H_2O$ are 0.01 wt. %, and for FeO and $CO_2$ are 0.1 wt. %. The concentration of the rare earth elements (REE) was determined by ICP-MS (PerkinElmer ELAN 9000 DRC-e) in the Institute of North Industrial Ecology Problems KSC RAS. Mineral abbreviations [41], and corresponding mineral names and formulas are shown in Table 2.

Additionally, the concentrations of rare earth elements were determined in 12 samples of alkaline rocks (5 samples of foidolites, 4 samples of foyaite, and 3 samples of

rischorrite) taken along the profile crossing the Main Ring of the Khibiny massif. Sampling points of alkaline rocks are shown in Figure 1b. The contents of the rock-forming oxides and the petrographic characteristics of these samples are presented in the works [24,42]. Here, we used data on the content of REE in alkaline rocks for comparison with aluminous hornfelses.

**Table 2.** Mineral abbreviations.

| Abbreviation [41] | Mineral | Formula* |
|---|---|---|
| Ab | albite | $Na(AlSi_3O_8)$ |
| Afs | alkali feldspar | $(K,Na)AlSi_3O_8$ |
| Alm | almandine | $Fe^{2+}_3Al_2(SiO_4)_3$ |
| And | andalusite | $Al_2SiO_5$ |
| Ann | annite | $KFe^{2+}_3(AlSi_3O_{10})(OH)_2$ |
| Arf | arfvedsonite | $NaNa_2(Fe^{2+}_4Fe^{3+})Si_8O_{22}(OH)_2$ |
| Ast | astrophyllite | $K_2NaFe^{2+}_7Ti_2(Si_4O_{12})_2O_2(OH)_4F$ |
| Crd | cordierite | $Mg_2Al_4Si_5O_{18}$ |
| Crn | corundum | $Al_2O_3$ |
| Fa | fayalite | $Fe^{2+}_2(SiO_4)$ |
| Fap | fluorapatite | $Ca_5(PO_4)_3F$ |
| Flr | fluorite | $CaF_2$ |
| Hc | hercynite | $Fe^{2+}Al_2O_4$ |
| Ilm | ilmenite | $Fe^{2+}Ti^{4+}O_3$ |
| Mag | magnetite | $Fe^{2+}Fe^{3+}_2O_4$ |
| Mnz-Ce | monazite-(Ce) | $Ce(PO_4)$ |
| Ms | muscovite | $KAl_2(Si_3Al)O_{10}(OH)_2$ |
| Nph | nepheline | $Na_3K(Al_4Si_4O_{16})$ |
| Pcl | pyrochlore-group mineral | $\mathbf{A_{2-m}\,B_2\,X_{6-w}\,Y_{1-n}}$<br>**A** = Na, Ca, Ag, Mn, Sr, Ba, Fe, Pb, Sn, Sb, Bi, Y, Ce, Sc, U, Th, □, or $H_2O$;<br>**B** = Ta, Nb, Ti, Sb, W;<br>**X** = O, OH, F;<br>**Y** = OH, F, O, □, $H_2O$, K, Cs, Rb. |
| Phl | phlogopite | $KMg_3(AlSi_3O_{10})(OH)_2$ |
| Pyh | pyrrhotite | $Fe_7S_8$ |
| Qz | quartz | $SiO_2$ |
| Rct | richterite | $Na(NaCa)Mg_5Si_8O_{22}(OH)_2$ |
| Sil | sillimanite | $Al_2SiO_5$ |
| Skn | sekaninaite | $Fe^{2+}_2Al_4Si_5O_{18}$ |
| Sps | spessartine | $Mn^{2+}_3Al_2(SiO_4)_3$ |
| Ttn | titanite | $CaTi(SiO_4)O$ |
| Uspl | ulvöspinel | $Fe^{2+}_2TiO_4$ |
| Zrn | zircon | $Zr(SiO_4)$ |

*—mineral formulas are given in accordance with IMA (International Mineralogical Association) list of minerals, with the exception of pyrochlore-group minerals.

## 4. Results

### 4.1. Petrography and Mineralogy of the Aluminous Hornfelses

The aluminous hornfels are fine-grained holocrystalline rocks, mostly characterized by honeycomb texture. The structure is massive or indistinctly banded due to the alternation of thin layers enriched or depleted in dark-colored minerals (Figure 4a). Because of the presence of a large number of segregations of dark-colored (annite,

hercynite, ilmenite, etc.) or light-colored (feldspars, fluorapatite, fluorite, etc.) minerals the structure of some varieties of hornfels is spotty (Figure 4b–d). The hornfelses are always crossed by numerous thin branchy veinlets (Figure 4c–e). Such veinlets are composed mainly of feldspar, and the other minerals are the same as in the surrounding hornfels. For example, in corundum-bearing hornfelses, one can find feldspar veins with large bright blue corundum crystals (Figure 4e). Moreover, large poikilitic crystals of nepheline (Figure 4f), sekaninaite, garnets, micas, and amphiboles are often found in hornfelses.

The rock-forming minerals of hornfels are feldspars, namely, (K,Na)-feldspar and albite, and hercynite, minerals of the annite-phlogopite and cordierite-sekaninaite series, fayalite, andalusite, sillimanite, quartz, corundum, muscovite, almandine, spessartine, nepheline, amphiboles (mainly arfvedsonite, ferro-nybøite, ferro-ferri-nybøite), and astrophyllite. The ratios of all the above minerals vary greatly even within the same xenolith and can be completely different in adjacent xenoliths. For example, a xenolith located on Eveslogchorr Mt. (Khibiny massif, sampling point 4, Figures 1b and 5a) has an oval shape and a size of 120 m × 40 m (Figure 5b). This xenolith consists of black, dark gray, sometimes with a greenish tint, banded, fine-grained hornfelses of various mineral compositions. The mineral associations that compose the xenolith on Eveslogchorr Mt. are presented in Figure 5c. In general, this xenolith can be considered representative, since it is composed of mineral associations most characteristic of aluminous hornfelses in both the Khibiny and Lovozero massifs.

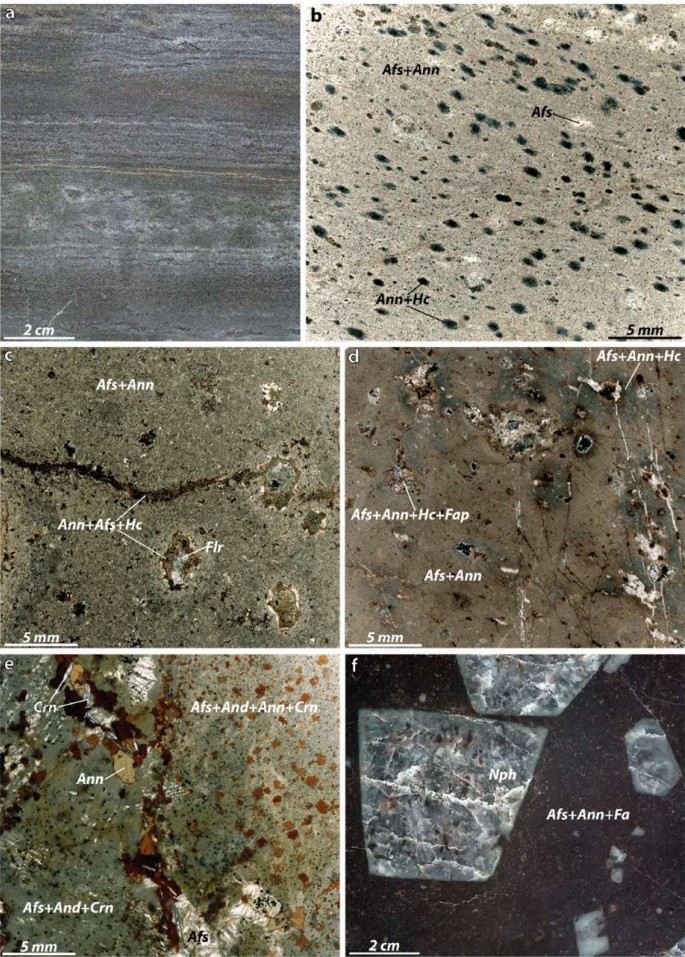

**Figure 4.** Typical textures of aluminous hornfelses. (**a**) hornfels with banded texture (sample KH-61-3); light- and dark-colored bands are due to the increased content of feldspar and annite +

fayalite, respectively; (**b**) hornfels with spotty texture (sample E-97-33); dark spots are annite + hercynite segregations and light spots are feldspar; (**c**) hornfels with zoned (fluorite in the center and annite + hercynite at the rim) segregations and thin annite-rich veins (sample LV-160/4); (**d**) hornfels with light- and dark-colored segregations connected by numerous feldspar veinlets (sample LV-121A); (**e**) veinlet composed of feldspar, annite, and corundum in corundum-andalusite-annite-feldspar hornfels (sample KCH-05-15); (**f**) large poikilitic nepheline crystals in fayalite-annite-feldspar hornfels (sample KH-31-1). (**a**,**f**)–photos of the polished samples surfaces; (**b**–**e**)–photos of thin sections in transmitted light.

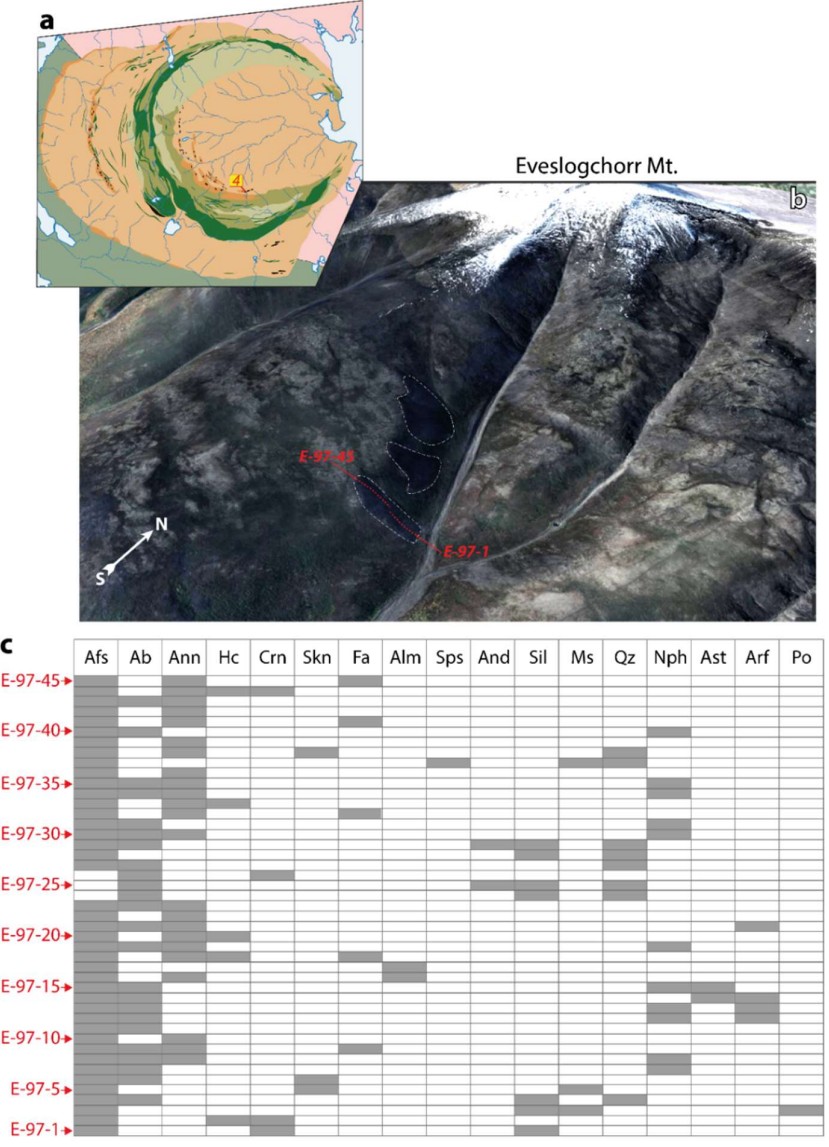

**Figure 5.** Outcrops of aluminous hornfelses on the southern slope of Eveslogchorr Mt. in the Khibiny massif. (**a**) location of xenoliths (sampling point 4) on the geological scheme of the Khibiny massif; (**b**) outcrops of three xenoliths (marked with a white dotted line). The sampling profile from point E-97-1 (67°39'59.6''N 33°58'56.5''E) to point E-97-45 (67°39'56.5''N 33°58'35.4''E) is shown with a red dotted line. The sampling interval is from 1.2 to 5 m; (**c**) rock-forming minerals (gray rectangles) of hornfelses sampling along the profile from point E-97-1 to point E-97-45. Only minerals whose content exceeds 10 mod. % are marked by gray rectangles.

We subdivided the rock-forming minerals of aluminous hornfelses into two groups: hornfels minerals and fenite minerals (Table 3). These groups differ in the chemical composition of the minerals and in the morphology of their grains. The hornfels minerals

are enriched in Al, $Fe^{2+}$, and K and form small polygonal grains, less often poikilitic crystals, whereas fenite minerals either replace the minerals of the hornfelses or form anhedral grains and large poikilitic crystals. The hornfelses minerals are very diverse and in turn are divided into three groups: Al-Si, Al-Fe, and 'ubiquitous' minerals. Minerals from the first and second groups almost never occur together, while minerals from the third group can be found both in association with minerals of the Al-Si group and in association with Al-Fe minerals.

**Table 3.** Groups of rock-forming minerals of aluminous hornfelses.

| Hornfels Minerals | | | Fenite Minerals |
|---|---|---|---|
| Al-Si minerals (high Si association) | ubiquitous minerals | Al-Fe minerals (low Si association) | |
| muscovite andalusite sillimanite quartz | (K,Na)-fieldspar albite ($Ab_{51-98}An_{2-48}Or_{0-1}$) sekaninaite-cordierite annite-phlogopite corundum almandine-spessartine | hercynite fayalite | nepheline albite ($Ab_{99-100}An_{0-1}$) alkali amphiboles astrophyllite titanite |

The main leucocratic minerals of the hornfelses in the Khibiny and Lovozero massifs are *(K,Na)-feldspar* and *albite*. Usually, the total content of these minerals in hornfelses is 75–90 vol.%; only in some muscovite- or sekaninaite-enriched lithologies does it decrease to 20–25%. Some of the studied samples are composed of homogeneous (K,Na)-feldspar grains with an $Ab_{32-75}Or_{25-68}$ composition (Figure 6a), but in most samples, feldspar is intensely albitized. The process of albitization begins with the formation of metasomatic perthite along the boundaries of the feldspar grains (Figure 6b). Metasomatic perthite is irregular and is mostly of vein- and flame-type. Such perthites consist of pure albite ($Ab_{99-100}$) and (K,Na)-feldspar, in which the sodium content is lower than in unaltered (K,Na)-feldspar (Figure 6b,c). The albite flames generally grow parallel to the normal perthite crystallographic plane, which is the orientation of the least lattice misfit between potassium feldspar and albite. With further development of the albitization, the volume of flame perthites increases (Figure 6d,e), and, finally, only relics of (K,Na)-feldspar in albite remain (Figure 6f). While pure albite ($Ab_{99-100}$) replaces feldspar, plagioclase $Ab_{51-98}An_{2-48}Or_{0-1}$ surrounds fluorite and/or fluorapatite grains (Figure 6g) and segregations of dark-colored minerals (Figure 6h). Representative chemical analyses of (K,Na)-feldspar and albite are shown in Supplementary Tables S1 and S2, respectively.

*Hercynite* is a characteristic rock-forming, less commonly accessory mineral of aluminous hornfelses in both the Khibiny and Lovozero massifs. The modal content of hercynite can reach 50%. Hercynite is commonly associated with feldspars, annite, fayalite, and ilmenite, as well as sekaninaite and corundum. The mineral forms rounded or anhedral grains that are either evenly distributed in the rock (Figure 7a–c) or form rounded segregations together with annite and ilmenite (Figure 7e,f). Representative chemical analyses of hercynite are shown in Supplementary Table S3. The compositions of hercynite from all studied samples are shown on the ternary $Fe^{2+}$–Mg–Zn diagram in Figure 8a. The content of Mg reaches 0.32 *apfu* (median 0.08 *apfu*), and Zn, 0.36 *apfu* (median 0.05 *apfu*). In addition, manganese is a typical impurity (up to 0.14 *apfu*).

In the hornfelses of the Khibiny and Lovozero massifs, Fe-Mg trioctahedral micas are one of the main rock-forming minerals. Minerals of the *annite–phlogopite* series form small plate crystals, aggregated in clusters (Figure 7e,f; Figure 9a) or evenly distributed in the rock (Figure 9c,d), as well as large (up to 8 mm across) grains (Figure 9b), with inclusions of feldspars, hercynite, ilmenite, zircon, monazite-(Ce), and other minerals. Micas intensively replace ilmenite and titaniferous magnetite (Figure 9d), and probably

also hercynite and fayalite, forming rims around the grains of these minerals. The ratio $Fe^{2+}/(Fe^{2+} + Mg)$ in the compositions of annite-phlogopite varies from 0.27 to 1 (Figure 8b). Annite is the most widespread, while phlogopite occurs much less frequently and mainly in hornfelses from the Lovozero massif. Mica consistently contains a significant amount of Ti (up to 0.40 *apfu*), Mn (up to 0.21 *apfu*), and F (up to 0.19 *apfu*), as well as a high content of Al, occupying both tetrahedral and octahedral (up to 0.49 *apfu*) sites in the crystal structure (Figure 8b). In addition to micas of the annite-phlogopite series, ***siderophyllite*** $KFe^{2+}_2Al(Si_2Al_2)O_{10}(OH)_2$ was also found in hercynite-enriched varieties of hornfelses. Representative microprobe analyses of trioctahedral micas are shown in Supplementary Table S4.

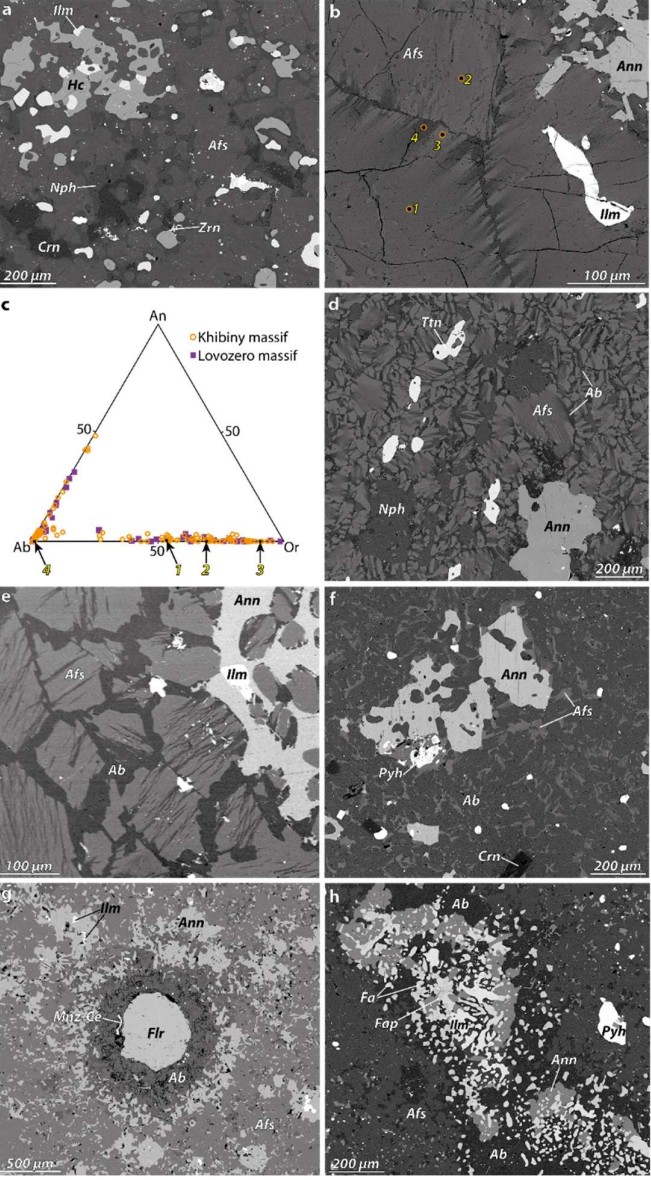

**Figure 6.** Morphology and chemical composition of feldspars from aluminous hornfels. (**a**) homogeneous (K,Na)-feldspar grains (sample M-01-2-8); (**b**) beginning of (K,Na)-feldspar albitization; flame perthites are formed at the grain boundaries (sample KCH-05-13); (**c**) albite (Ab)–orthoclase (Or)–anorthite (An) ternary feldspar diagram. The points with numbers (1–4) correspond to the points in Figure 6b; (**d**) intensely albitized (K,Na)-feldspar (sample LV-122); (**e**), (**f**) (K,Na)-feldspar relics in albite (samples KH-61-11 and KCH-05-31, respectively); (**g**) albite (Ab$_{51}$An$_{48}$Or$_1$)

rims around a fluorite grain (sample M-01-2-4); (**h**) albite ($Ab_{58}An_{42}$) rims around the segregations of dark-colored (annite+ilmenite) minerals (sample M-01-3). BSE-images (**a**,**b**,**d**–**h**).

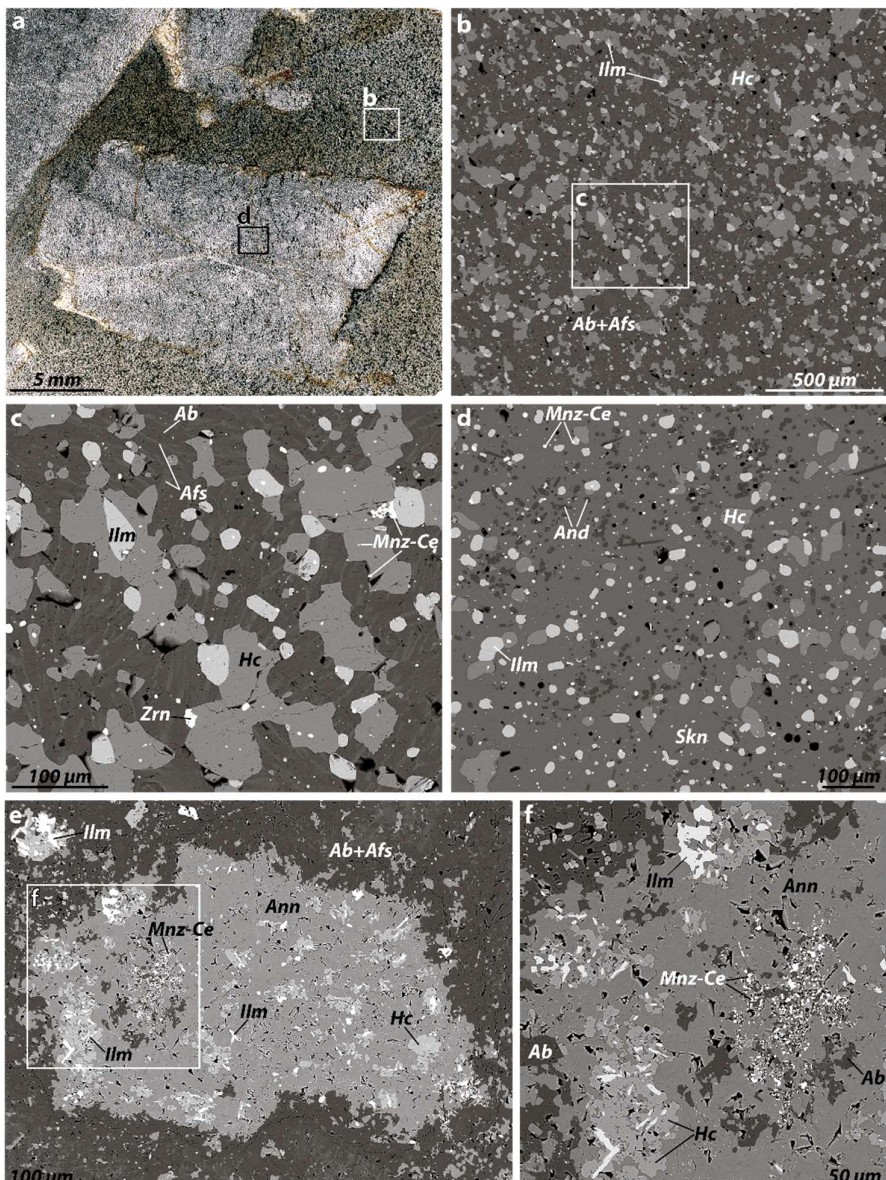

**Figure 7.** Mineral associations of aluminous hornfelses. (**a**) large poikilitic crystals of sekaninaite in a fine-grained mass of feldspars, annite, and hercynite (sample KCH-4); (**b**) detailed fragment of Figure 7a; (**c**) detailed fragment of Figure 7b; hercynite grains in association with feldspars, ilmenite, zircon, and monazite-(Ce); (**d**) fragment of the poikilitic crystal of sekaninaite with small inclusions of the hercynite, ilmenite, andalusite, and monazite-(Ce); (**e**) hercynite+ilmenite+annite segregation in hornfels (sample LV-119); (**f**) detailed fragment of Figure 7e. Photo of polish section in transmitted light (**a**) and BSE-images (**b**–**f**).

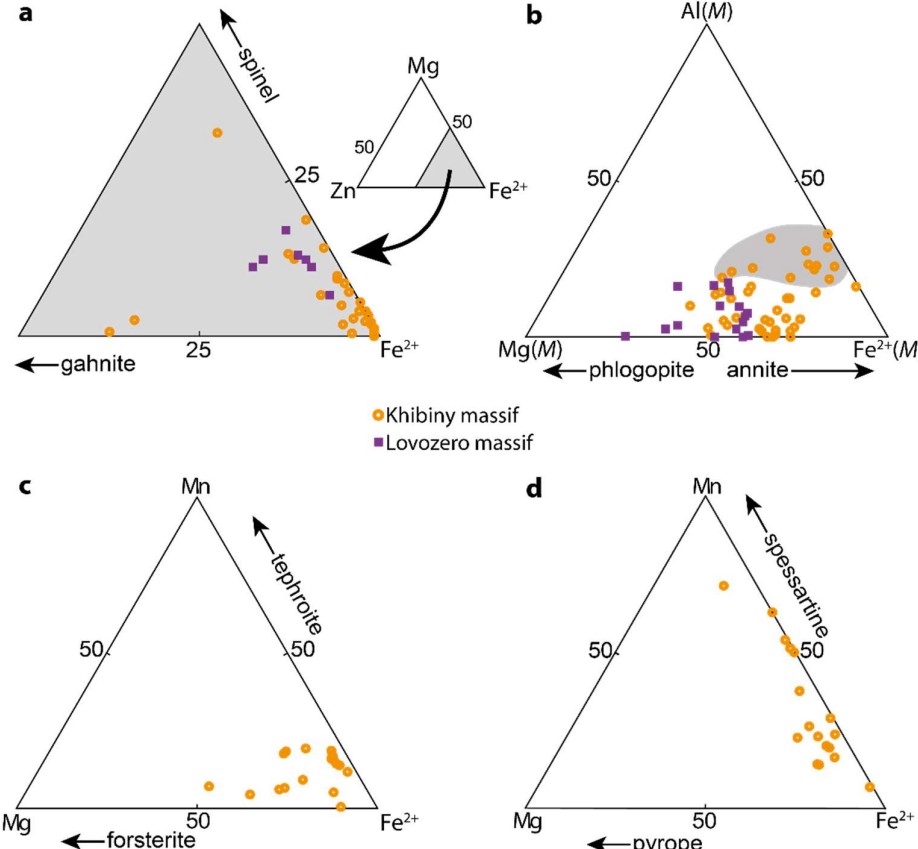

**Figure 8.** Compositions of rock-forming minerals from aluminous hornfelses. (**a**) $Fe^{2+}$–Mg–Zn diagram showing the hercynite compositions; (**b**) diagram showing the ratio of cations in *M* site in the crystal structure of trioctahedral micas. General formula of micas $I\ M_{2\text{-}3}\ \square_{1\text{-}0}\ T_4\ O_{10}\ A_2$ [43]. The compositions corresponding to siderophyllite are shown on a gray background; (**c**) $Fe^{2+}$–Mn–Mg diagram showing the fayalite compositions; (**d**) $Fe^{2+}$–Mn–Mg diagram showing the garnet compositions.

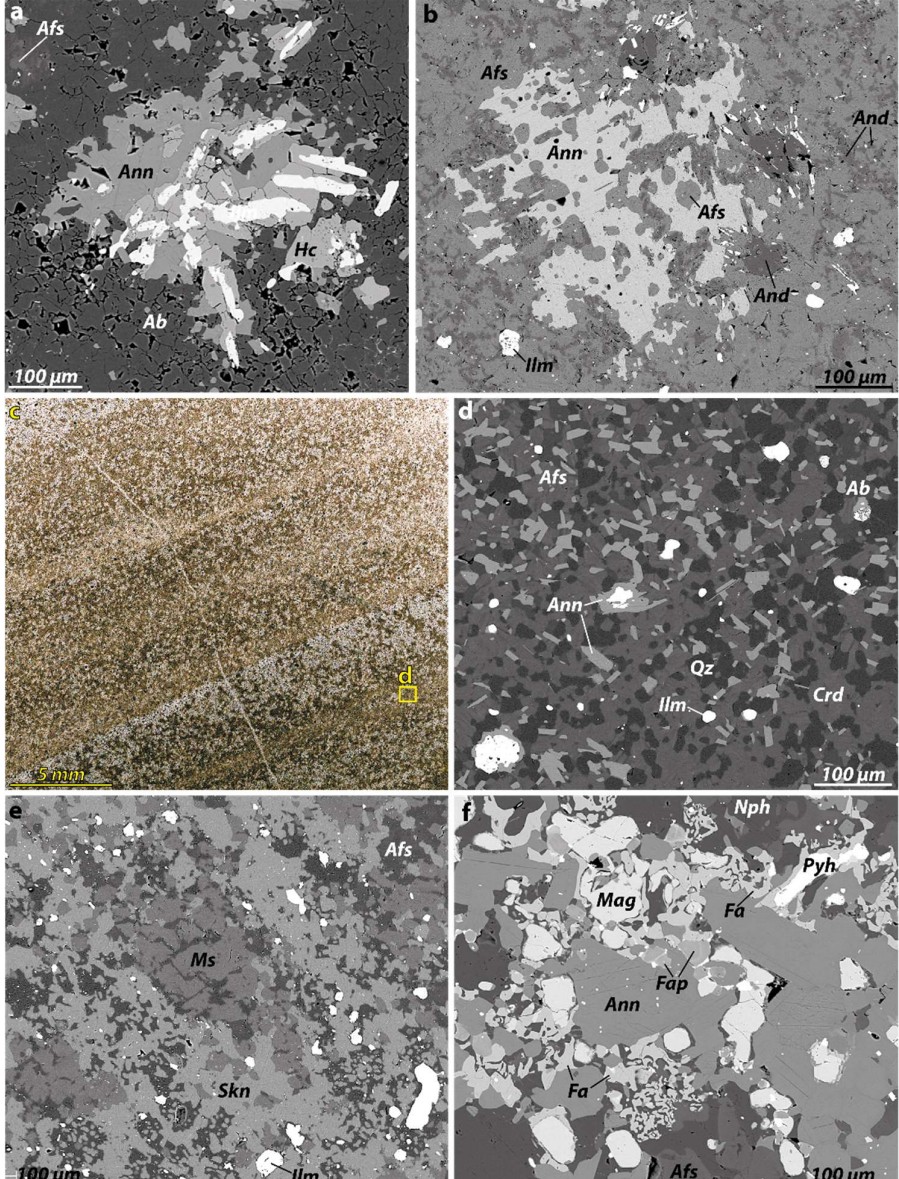

**Figure 9.** Mineral associations of hornfelses. (**a**) plate annite crystals and fine albite grains surround ilmenite and hercynite grains (sample LV-119); (**b**) poikilitic annite crystal with inclusions of (K,Na)-feldspar and andalusite (sample KCH-05-15); (**c**) hornfels with banded texture where light-colored layers enriched in quartz, and dark-colored ones enriched in annite and cordierite (sample LV-01-45); (**d**) detailed image of Figure 9d; annite replaces ilmenite and titaniferous magnetite; (**e**) hornfels consisting of sekaninaite, muscovite and alkali feldspar (sample E-97-5); (**f**) fayalite in association with annite, titaniferous magnetite and fluorapatite (sample E-97-41). Photo of polish section in transmitted light (**c**) and BSE-images (**a**,**b**,**d**–**f**).

The minerals of the ***cordierite–sekaninaite*** series form both large poikilitic crystals with numerous small inclusions of hercynite, ilmenite, andalusite, feldspars, pyrrhotite, annite, quartz, and monazite-(Ce) (Figure 7a,d), and small rounded or anhedral grains (Figure 9c,d). Cordierite occurs much less frequently and mainly in hornfelses from the Lovozero massif. Chemical analyses of the minerals of the cordierite-sekaninaite series are presented in Supplementary Table S5. Characteristic impurities in sekaninaite and cordierite are Mn (up to 0.30 *apfu*) and Na (up to 0.11 *apfu*).

*Fayalite* is a typical accessory or rock-forming mineral of hornfelses from xenoliths within the Khibiny massif and is usually found in close association with feldspars, annite, ilmenite, titaniferous magnetite, and fluorapatite (Figures 6h and 9f). In some lithologies, the fayalite content reaches 30 mod. %. This mineral forms small rounded or anhedral grains up to 0.3 mm across, which are usually not evenly distributed in the rock but form clusters together with other dark-colored minerals. The ratio $Fe^{2+}/(Fe^{2+}+Mg)$ in the fayalite varies from 0.54 to 0.97; typical impurities are Mn (up to 0.38 *apfu*), Ca (up to 0.02 *apfu*), and Ti (up to 0.01 *apfu*). Representative chemical analyses of the fayalite are accessible in Supplementary Table S6, and Figure 8c shows the compositions of fayalite in the $Fe^{2+}$–Mn–Mg diagram.

Garnets of the *almandine–spessartine* series, like fayalite, are characteristic accessory minerals of aluminous hornfelses of the Khibiny massif, while these minerals were not found in the Lovozero hornfelses. In some Khibiny lithologies, these garnets are rock-forming minerals (Figure 5c), and their content reaches 15 mod. %. Almandine is predominantly associated with hercynite, annite, sekaninaite, and fayalite, while spessartine is associated with muscovite and quartz. The garnets form large (up to 1.2 cm across) poikilitic crystals with inclusions of surrounding minerals. The representative chemical analyses of the minerals of the almandine-spessartine series are presented in Supplementary Table S7. The ratio of Mn and $Fe^{2+}$ varies widely (Figure 8b), and typical impurities are magnesium (up to 0.39 *apfu*) and calcium (up to 0.08 *apfu*).

*Muscovite* in aluminous hornfelses forms small (50 μm on average) plate grains (Figure 9e), as well as relatively large (up to 3 mm in diameter) poikilitic crystals in association with feldspars, quartz, sillimanite, andalusite and sekaninaite. The main impurities in muscovite are Na (up to 0.12 *apfu*) and $Fe^{3+}$ (up to 0.12 *apfu*), replacing K and Al, respectively (Supplementary Table S8). *Corundum* forms rounded grains (up to 1 mm across), uniformly dispersed in the main fine-grained mass of hornfelses (Figure 4e, 6a) or long-prismatic crystals (Figure 4e). Iron (up to 0.96 wt. % $Fe_2O_3$), Si (up to 0.45 wt. % $SiO_2$), and Ti (up to 0.33 wt. % $TiO_2$) are common impurities in the chemical composition of corundum. *Sillimanite* occurs as colorless needle-shaped crystals (up to 2 mm in length), as well as radiated aggregates (up to 1 mm in diameter). *Andalusite* is found as inclusions in poikilitic crystals of sekaninaite (Figure 7d), and small (up to 0.5 mm across), pale pink grains in association with feldspars, annite, muscovite, and corundum (Figures 4e and 9b); it also forms aggregates of small grains and, more rarely, well-formed prismatic crystals. In the same xenolith, samples were found containing both sillimanite in association with andalusite and samples containing only sillimanite or only andalusite. Perhaps this is due to the fact that the P-T conditions for the formation of aluminous hornfelses were close to the sillimanite-andalusite equilibrium. According to microprobe analysis, for andalusite, as well as for sillimanite, a slight admixture of iron is characteristic (up to 0.02 *apfu*).

*Nepheline* forms anhedral grains filling feldspar and albite interstices (Figure 6a), and poikilitic crystals (Figure 4f) with numerous inclusions of the surrounding minerals. This mineral constantly contains an admixture of iron (up to 0.11 *apfu*). *Quartz* in aluminous hornfelses is probably a relic mineral. In the Khibiny massif, we found quartzite, where (K,Na)-feldspar forms thin rims around quartz grains and fills interstices, and, in addition, small grains of sekaninaite are located along the boundaries of quartz and (K,Na)-feldspar (Figure 10a). The same xenolith contained poikilitic annite crystals and intergrowths of fayalite with sekaninaite, while a (K,Na)-feldspar rim is always present between all these minerals and quartz (Figure 10b). In some varieties of aluminous hornfelses (for example, in cordierite- or sekaninaite-bearing hornfelses), the layering inherited from the protolith is probably preserved, i.e., the alternation of thin layers with different quartz contents (Figure 9c). In addition to sekaninaite-cordierite, quartz is also associated with sillimanite, andalusite, muscovite, and garnets. *Amphiboles* in aluminous hornfelses are very diverse; they occur in any association and usually form large poikilitic grains (Figure 10c) with inclusions of surrounding minerals. In the studied samples, arfvedsonite is most common,

but ferro-nybøite and ferro-ferri-nybøite, as well as ferro-pargasite, edenite, richterite, and ferri-katophorite have also been found. All varieties of amphiboles contain fluorine in their chemical composition (from 0.04 to 0.70 *apfu*).

The typical accessory minerals of aluminous hornfels in Khibiny and Lovozero massifs are fluorapatite, fluorite, monazite-(Ce), titanite, ilmenite, titaniferous magnetite, ulvöspinel, pyrrhotite, and zircon.

***Fluorapatite*** and ***fluorite*** occur in all varieties of aluminous hornfelses of both the Khibiny and Lovozero massifs. These minerals form small grains evenly distributed in the rock (Figure 9f), as well as clusters of small grains (Figure 10e) or relatively large grains surrounded by Ca-rich albite, annite, and ilmenite (Figure 6g). Fluorapatite contains an admixture of Sr (up to 1.41 *apfu*) and Na (up to 0.53 *apfu*) and is enriched in rare earth elements (total content of REE up to 0.38 *apfu*). Usually, fluorapatite grains are zoned, with calcium core and rims enriched in Sr, Na, and REE. However, the main carrier of REE in aluminous hornfelses is ***monazite-(Ce)***. This mineral is exclusively widespread and forms dissemination of very small (up to 10 μm) rounded grains (Figure 10f,g) in all varieties of hornfelses. Rare earth elements are also present in the composition of accessory titanite, which, like amphiboles, forms poikilitic grains or crystals (Figure 10d). The total *REE* concentration in titanite is low and never exceeds 0.03 *apfu*.

Among the Fe-Ti oxides, ilmenite is the most common, titaniferous magnetite is less widespread, and ulvöspinel is even rarer. Grains of titaniferous magnetite can be either homogeneous or contain thin lamellae of ilmenite and/or ulvöspinel. Similarly, ulvöspinel grains can be either completely homogeneous or contain thin exsolution lamellae of ilmenite (Figure 10g). Pyrrhotite forms extremely small inclusions in the rock-forming minerals (mainly in feldspars) of aluminous hornfelses. Pyrrhotite is often replaced by pyrite, marcasite, and goethite, which is why xenoliths often have an orange-brown color on the day surface (Figure 2b,c).

Other accessory minerals of aluminous hornfelses are xenotime-(Y), titanite, minerals of the pyrochlore group, fluocerite, loparite-(Ce), zirconolite, chevkinite-(Ce), topaz, sphalerite, molybdenite, graphite, native iron (Figure 10h), and troilite.

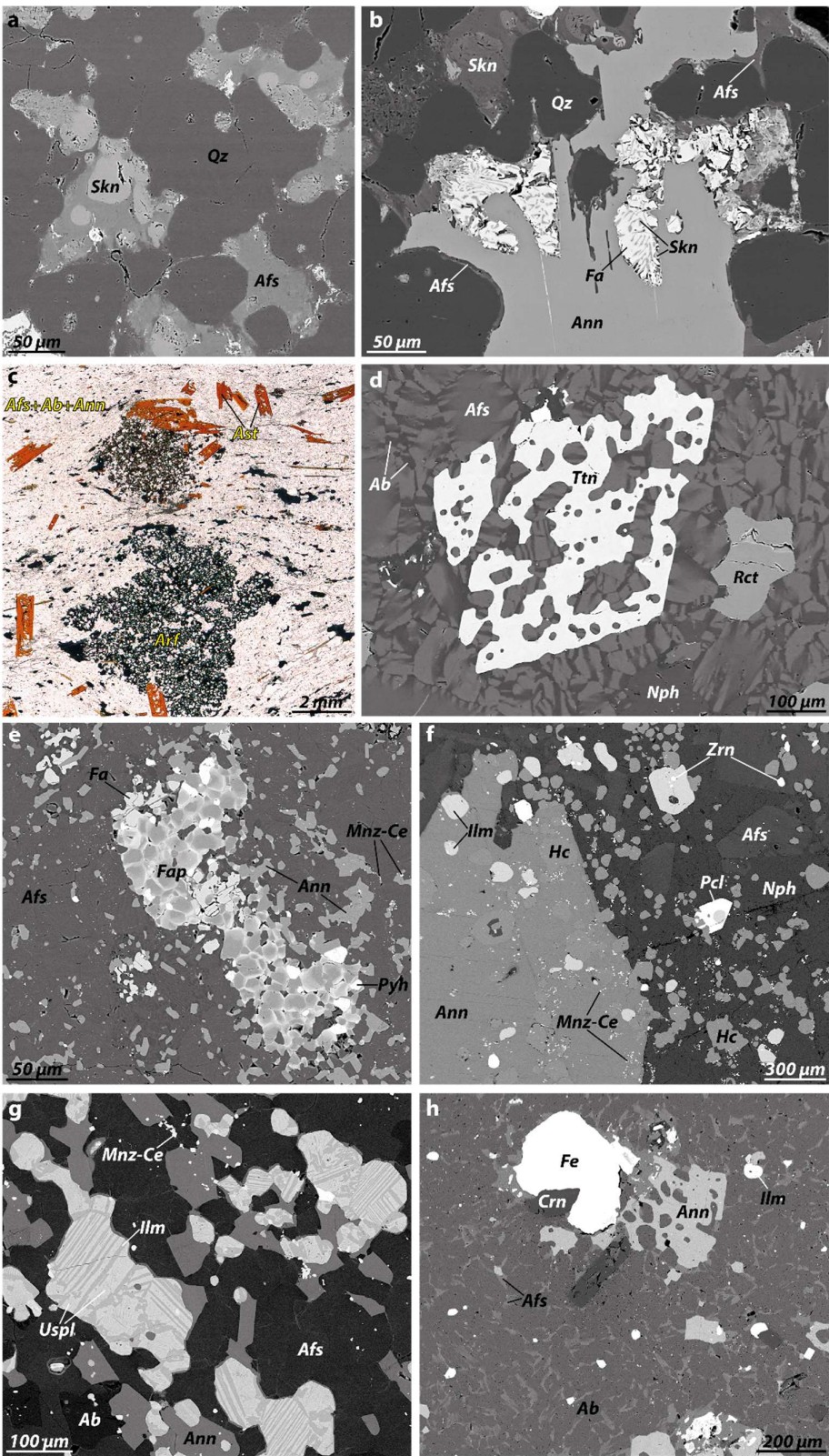

**Figure 10.** Mineral associations of quartzite and aluminous hornfelses. (**a**) sekaninaite grains located along the boundaries of quartz and (K,Na)-feldspar (quartzite, sample KH-110); (**b**) fayalite-sekaninaite intergrowths in annite grain surrounded by (K,Na)-feldspar rim (quartzite, sample KH-111); (**c**) large poikilitic grains of arfvedsonite and astrophyllite among (K,Na)-feldspar (sample E-

97-14); (**d**) poikilitic titanite crystal in fine-grained feldspar mass (sample LV-122); (**e**) zonal fluorapatite grains in fayalite-annite-feldspar hornfels (sample KH-61-6); (**f**) small grains of monazite-(Ce), and crystal of the pyrochlore-group mineral in nephelinized hercynite-annite-feldspar hornfels (sample M-01-1-10); (**g**) ulvöspinel grains with ilmenite lamellae in annite-feldspar hornfels (sample M-01-2-12); (**h**) native iron in corundum-annite-feldspar hornfels (sample KCH-05-31). BSE-images (**a**,**b**,**d**–**h**) and photo in transmitted light (**c**).

### 4.2. Rock Chemistry

Data on the content of major oxides and volatiles in aluminous hornfelses (from sample E-97-1 to sample E-97-45, Figure 5b,c) are presented in Supplementary Table S9 and are shown as bar diagrams in Figure 11. This figure also displays the contents of major components in Devonian volcaniclastic rocks, which are presumably protoliths of hornfelses. Petrography investigations of unaltered volcaniclastic rocks in the Lovozero massif have shown that the transition from basalt (and basaltic tuff) to tuffite and quartzite occurs exclusively due to the successive "dilution" of basalts/basalts tuff with quartz [32]. Assuming the minimum content of $SiO_2$ and the maximum concentration of all other components in unaltered basalt/basalt tuff, we can represent the composition of volcaniclastic rocks as gradients. Thus, it is possible to compare the composition of such a complex protolith with the distribution of components in aluminous hornfelses. In addition, Figure 11 shows the content of the main components in the foidolites and foyaite of the Khibiny massif according to the data of Ivanuyk and co-authors [42] and the chemical composition of greenstone belt rocks according to Gorstka [14].

The concentrations of major oxides in aluminous hornfelses vary widely. The $SiO_2$ content (median 53.35 wt. %) varies from 14.77 wt. % in oxide-enriched associations (corundum-, hercynite-, muscovite-, sillimanite-bearing hornfelses) to 59.77 wt. % in associations with a high content of feldspars and quartz. The $Al_2O_3$ content (median 20.61 wt. %), on the contrary, is maximum in hornfelses containing rock-forming aluminum oxides or silicates with a high Al/Si ratio, such as sillimanite or muscovite. The concentration of $TiO_2$ varies from 0.68 to 3.92 wt. % (median 1.66 wt. %) and mainly depends on the modal content of ilmenite and titaniferous magnetite, as well as annite, which always contains a significant admixture of titanium (Table S4). The $FeO/Fe_2O_3$ ratio in aluminous hornfelses reaches 12.7 due to the presence of a large number of minerals containing only ferrous iron (annite, hercynite, fayalite, sekaninaite). The content of ferrous iron in hornfelses varies quite widely (0.92–15.70 wt. % FeO), which is related to a large diversity of rock-forming, iron-bearing minerals and variations in the modal content of these minerals. Ferric iron (median 2.01 wt. %) is included in the less widespread titaniferous magnetite and amphiboles.

The main role of Mn and Mg in aluminous hornfelses is the isomorphic substitution of ferrous iron. Manganese is an important impurity in ilmenite, fayalite, hercynite, garnets, and the main carriers of Mg are minerals of the annite-phlogopite and sekaninaite-cordierite series, as well as amphiboles. Accordingly, the concentrations of magnesium and manganese in hornfelses are quite low: 0.07–2.49 (median 0.49) wt. % MgO and 0.01–2.64 (median 0.36) wt. % MnO.

The distributions of $Na_2O$ and $K_2O$ are bimodal. The maximum concentrations of $Na_2O$ were found in hornfelses enriched in albite, nepheline, and alkaline amphiboles, and the maximum concentrations of $K_2O$ are associated primarily with the increased content of (K,Na)-feldspar. The main calcium carriers are fluorapatite and fluorite, as well as, to a lesser extent, Ca-enriched albite. The content of CaO in aluminous hornfelses varies from 0.20 to 3.13 wt. %, and the median content is 0.56 wt. %. Aluminous hornfelses are significantly enriched in fluorine, the main carriers of which are micas of the annite-phlogopite series, fluorapatite, fluorite, as well as amphiboles. Respectively, the fluorine content in aluminous hornfelses can reach 1.63 wt. % (median 0.14), while the average fluorine content in foyaite is 0.07 wt. % [42]. In terms of fluorine content, hornfelses are closer to rischorrite, in which the average content of this element is 0.12 wt. % [42].

In addition, aluminous hornfelses are significantly enriched in sulfur, mainly due to the high content of pyrrhotite, which is present in all varieties of these rocks. The content of $S_{tot}$ in hornfelses (median 0.75 wt. %) exceeds that in foyaite (average 0.04 wt. %) and foidolites (average 0.07 wt. %) [42]. The chlorine concentration in aluminous hornfelses (median 0.01 wt. %), on the contrary, is lower than in foyaite and foidolites (0.08 and 0.05 wt. %, respectively) [42].

The main carriers of REE in aluminous hornfelses are the ubiquitous monazite-(Ce), fluorapatite, and titanite, as well as less widespread pyrochlore-group minerals, fluocerite-(Ce), fergusonite-(Y), xenotime-(Y), zirconolite, and other minerals. The total content of $REE_2O_3$ in the chemical composition of the studied samples reaches 0.47 wt. %, and the median content is 0.09 wt. %, while the $REE_2O_3$ in the foyaite of the Khibiny massif is 0.17 wt. % (Figure 12a) [44].

Figure 12b shows the chondrite-normalized REE spectra in aluminous hornfelses, which differ greatly in mineral composition, as well as chondrite-normalized REE spectra in quartzite (sample KH-111) and the main types of alkaline rocks of the Khibiny massif. The $(La/Lu)_{cn}$ ratio (where «cn» mean chondrite-normalized) in hornfelses varies from 25.84 to 29.20, in quartzite $(La/Lu)_{cn} = 19.78$, and in rischorrite, this ratio varies from 18.53 to 32.23. In foyaite and foidolites, the $(La/Lu)_{cn}$ ratio varies from 44.88 to 65.17 and from 28.33 to 67.33, respectively (Supplementary Table S10).

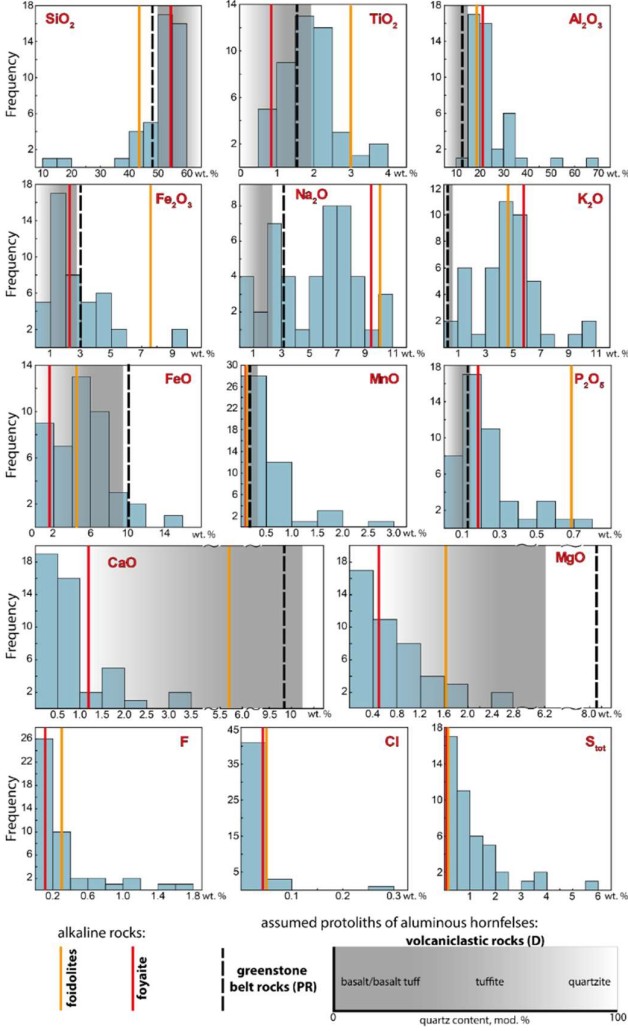

**Figure 11.** Comparison of the chemical compositions of aluminous hornfelses, volcanoclastic rocks, greenstone belt rocks, and alkaline rocks, namely foyaite and foidolites (Khibiny massif). The

compositions of aluminous hornfelses are presented as bar diagrams, and the compositions of volcaniclastic rocks [32] are shown as gradients. Red and yellow lines show the average composition of foyaite and foidolites [42], respectively. Dotted lines show the composition of greenstone rocks near the contact with the Khibiny massif according to [14].

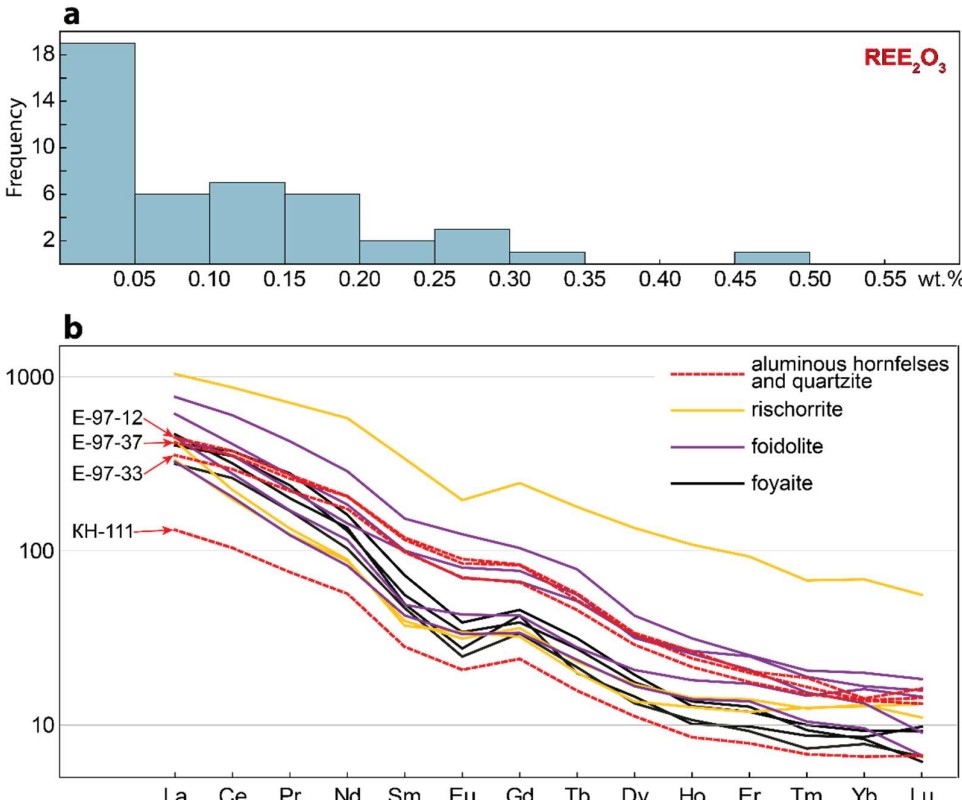

**Figure 12.** Rare earth elements in aluminous hornfelses and alkaline rocks of the Khibiny massif. (**a**) bar diagram showing the distribution of $REE_2O_3$ content in hornfelses; (**b**) chondrite-normalized REE patterns in aluminous hornfelses and main types of alkaline rocks of the Khibiny massif. Sample numbers correspond to those in Figure 5c, where sample E-97-12 consists of (K,Na)-feldspar, albite, nepheline, and arfvedsonite; sample E-97-33 consists of (K,Na)-feldspar, annite, hercynite; sample E-97-37 consists of (K,Na)-feldspar, spessartine, muscovite, and quartz. Chondrite values are from [45].

## 5. Discussion

Mineral associations observed in xenoliths within the Khibiny and Lovozero massifs are characteristic of high-grade metapelitic rocks of predominantly sedimentary origin. Hercynite-gahnite-rich spinels are found in such rocks in association with garnet, quartz, $Al_2SiO_5$ polymorphs, corundum, cordierite, and other phases [46–48]. In addition, mineral associations including hercynite, annite, and muscovite can be formed as a result of the fenitization of pelitic rocks under the influence of alkaline plutons [49]. However, among the country rocks of both the Khibiny and Lovozero massifs, neither aluminous sediments nor metapelites were found [14,35]. However, even if we assume that aluminous hornfelses were formed during the metasomatism of the kyanite schists, it remains unclear why the xenoliths of the kyanite schists were located precisely near foidolites and nowhere else.

Let us assume that the protolith of aluminous hornfelses was either the rocks surrounding the massifs (Proterozoic greenstone rocks, Archean gneisses) or the roof rocks (Devonian volcaniclastic rocks). The study of the mineralogy of the samples along the profile crossing the xenolith on Eveslogchorr Mt. (Figure 5) showed that the protolith

of the hornfelses was very variable in mineral composition, i.e., it probably represented an interbedding of chemically contrasting rocks. The Archean rocks near the contact with the Khibiny and Lovozero massifs are represented by homogeneous garnet-biotite gneisses and migmatites with a very uniform chemical composition [14]. In addition, unaltered/weakly fenitized xenoliths of these rocks are very rare within the Khibiny and Lovozero massifs, in contrast to the weakly fenitized xenoliths of Devonian and Proterozoic rocks. Therefore, we can assume that Devonian and/or Proterozoic rocks were the most probable protoliths of the aluminous hornfelses.

The distribution maxima of $TiO_2$, $MnO$, and $P_2O_5$ in aluminous hornfelses coincide with the content of each of the components in Devonian basalt/basalt tuff, and in tuffite, the contents become lower than the distribution maxima (Figure 11). Consequently, during the possible metasomatic alteration of basalts and basalt tuff, there was no gain or loss of the abovementioned components, and during the alteration of the tuffite, there was an insignificant gain. The concentrations of $Fe_2O_3$, $FeO$, $MgO$, and $CaO$ in volcaniclastic rocks are higher than the distribution maxima in hornfelses, which suggests the loss of these components. Moreover, the losses of the above-mentioned components were most intense in basalt/basalt tuff. Indeed, in the Khibiny massif, nepheline syenites adjacent to xenoliths of aluminous hornfelses (Figure 3b) contain rock-forming clinopyroxenes enriched in Ca, Mg, and $Fe^{2+}$ (diopside and hedenbergite end-members) [42]. In addition, nepheline syenites near contact with aluminous hornfelses are enriched in titanite [50]. The Lovozero massif differs from the Khibiny in its extremely low calcium content [17]; therefore, titanite, as well as Ca-enriched clinopyroxenes and amphiboles, are not characteristic of the rocks of the massif. Fluorapatite occurs mainly in urtite and contains a significant amount of impurities (Sr, REE) that replace calcium [17]. However, xenoliths of aluminous hornfelses in the Lovozero massif are within the apatite-titanite ore occurrence (Figure 3a). It is possible that the formation of this mineralization is associated with the removal of calcium during a metasomatic alteration in the calcium-rich protolith.

The distribution maximum of $SiO_2$ in aluminous hornfelses corresponds with the content of $SiO_2$ in basalts/basalt tuffs and tuffite, which shows low mobility of this component during metasomatism. The distribution maximum of aluminum in hornfelses is higher than the content of this component in volcaniclastic rocks, which indicates the gain of Al during metasomatic alteration. The bimodal distribution of $Na_2O$ and $K_2O$ indicates a significant gain of these components during the transformation of volcaniclastic rocks into aluminous hornfelses. Thus, if Devonian volcaniclastic rocks really were the protolith of aluminous hornfelses, then the process of metasomatic alteration consisted of the addition of $Na_2O$, $K_2O$, and $Al_2O_3$ and the removal of $FeO$, $Fe_2O_3$, $MgO$, and $CaO$.

Assuming that greenstone rocks were the protolith of aluminous hornfelses, the process of metasomatic alteration was similar and involved the gain of $Na_2O$, $K_2O$, and $Al_2O_3$ and the loss of $FeO$, $Fe_2O_3$, $MgO$, and $CaO$. In any case, the composition of hornfelses approached that of alkaline rocks, since the distribution maxima of $SiO_2$, $Na_2O$, $K_2O$, and $Al_2O_3$ in hornfelses are close to the content of these components in foyaite and foidolites.

According to the studies of Gorstka [14] and Tikhonenkova [31], during the fenitization of both Archean gneisses at the contact of the Khibiny and Lovozero massifs, alkalis were intensively added, but the aluminum remained immobile. However, during the metasomatic alterations of xenoliths within the massif, besides the gain of sodium and potassium, there was the essential addition of aluminum. Probably, the reason is that the rocks surrounding the massif were influenced by fluids exsolved from the nepheline-syenitic melt, while xenoliths inside the Khibiny massif were affected by fluids separated from the foidolitic melt. That is why aluminous hornfelses are located exclusively near foidolite bodies. In the Khibiny massif, aluminous hornfelses are located mainly near the contact of rischorrite and foyaite (Figure 3b). According to previous works [24,27,51], rischorrite (and lyavochorrite) in the Khibiny massif were formed as a result of the metasomatic influence of foidolites intruded along a Main Ring fault on early crystallized

foyaite. In this process, $Na_2O$, $K_2O$, $Al_2O_3$, $P_2O_5$, F, Cl, and $H_2O$ were added to foyaite. Indeed, the contacts between rischorrite and foyaite are gradual, and a large number of hydrothermal veins near this contact (Figure 3b) indicates a significant role of fluids in this zone.

We assume that the main reason for the formation of Al-enriched hornfelses is the high mobility of aluminum in the fluids expelled from cooling crystallizing foidolites. Fluorine can dramatically increase the mobility of Al. For example, the experiments of Tagirov et al. [52] at 400 °C and 500 bars demonstrate that when the F concentration in an aqueous fluid reaches $10^{-4}$ m (~2 ppm), the Al concentration from the dissolution of the Al silicates is 1.5 times higher than in pure water; at $F = 10^{-3}$ m, the Al concentration is ~6 times higher. According to [52], temperature increases and pressure decreases promote the formation of Na-Al-OH-F complexes with an increase in $NaAl(OH)_2F^0_{2(aq)}$ stability relative to that of $NaAl(OH)_3F^0_{(aq)}$. It follows that the mobility of Al is significant in fluorine-containing high-temperature liquids with low density. In the Khibiny massif, foidolites and (titanite-)apatite-nepheline contain the largest concentrations of fluorine. Aluminous hornfelses and rischorrite rank second in fluorine content [11,42]. In hornfelses, fluorine is a part of both rock-forming (for example, micas and amphiboles) and accessory (fluorapatite, fluorite, topaz, cryolite, etc.) minerals. So, we propose that the reason for the formation of aluminous hornfelses was the influence on the protolith of fluids expelled from crystallizing foidolites and containing, in addition to alkalis, aluminum in the form of complexes with fluorine.

In fact, aluminous hornfelses are alkaline metasomatites (fenites). The presence of fluorine in the fluid is the key factor determining the high mobility of aluminum and, consequently, the formation of aluminous mineral associations. If the fluid were not enriched in fluorine, instead of aluminous hornfelses, classical fenites (rocks of alkali-syenitic composition) would form, consisting of aegirine, alkaline amphiboles, feldspar, albite, and nepheline.

When a fluid containing Na-Al-OH-F complexes is affected on basic rocks such as basalts or basalt tuffs, the sodium activity in the fluid is buffered by a reaction similar to that for the albitization of calcic plagioclase [53]:

$$CaAl_2Si_2O_8 + 2Na^+ + 4SiO_2 = 2NaAlSi_3O_8 + Ca^{2+}. \tag{1}$$

In the Khibiny and Lovozero case, the alteration probably had the following form:

$$CaAl_2Si_2O_8 + Na^+ + K^+ + 4SiO_2 = 2(K,Na)AlSi_3O_8 + Ca^{2+}. \tag{2}$$

(K,Na)-feldspar is silica-rich compared with oligoclase-andesine, but the concentrations of $SiO_2$ in basalts and aluminous hornfelses are similar (Figure 11). Therefore, it can be assumed that during the metasomatic alteration of basalts/basalt tuffs (in general, any mafic rocks), almost all the $SiO_2$ was spent on (K,Na)-feldspar crystallization.

As a result, the fluid gained calcium, and the *a*F(activity of fluorine) in the fluid was buffered by fluorite or/and fluorapatite precipitation. Clusters of Ca-rich minerals are also often formed (Figure 6g). Due to the breakdown of the Na-Al-OH-F complexes, aluminum precipitated in the form of micas (annite-phlogopite and siderophyllite), which, as we assume, replaced clinopyroxenes and olivine of the putative protolith. Extremely high aluminum contents in micas (annite, siderophyllite) indicate a significant excess of aluminum during metasomatic alterations. A high aluminum content (combined with a deficiency of $SiO_2$) has been responsible for the crystallization of aluminum oxides such as corundum and hercynite. The widespread hercynite-annite-feldspar (Figures 4b and 7e), fayalite-hercynite-annite-feldspar, and hercynite-corundum-feldspar (Figure 6a) hornfelses, i.e., rocks containing, in addition to feldspar, aluminum oxides, and silicates with a minimum content of Si, were formed in this way.

When a hydrothermal fluid containing Na-Al-OH-F complexes affected quartz-enriched rocks, such as tuffite, quartz and calcic plagioclase were intensively replaced by

(K,Na)-feldspar (Figure 10a,b). The gained aluminum, as in the case of the alteration of basalts/basaltic tuffs, was included in the composition of aluminous minerals, but the content of silicon in these minerals was higher than in annite (ideally 35.15% $SiO_2$) and fayalite (ideally 29.49% $SiO_2$). The reason is the higher content of Si in the protolith. As a result, muscovite (ideally 45.21% $SiO_2$) and sekaninaite (ideally 47.52% $SiO_2$) are formed upon the metasomatic alteration of quartz-enriched lithologies. There was no such extreme Si deficiency as in the alteration of basalts/basalt tuffs in this case; therefore, andalusite and sillimanite are formed instead of corundum in association with muscovite. The feldspar-muscovite-sekaninaite (Figure 9e), sillimanite-muscovite-feldspar hornfelses, i.e., rocks containing, in addition to feldspar, silicates with a low silica content, were formed by this way.

The formation of hornfelses associations was followed by the crystallization of typical fenite minerals. (K,Na)-feldspar was intensely albitized (Figure 6b,d–f). Moreover, anhedral grains (Figure 6a) and poikilitic crystals of nepheline (Figure 4f), as well as alkali amphiboles, astrophyllite (Figure 10c), and titanite (Figure 10d) were formed.

As shown above, the $(La/Lu)_{cn}$ ratio in hornfelses is very close to that in rischorrite and much lower than in foyaite and foidolites. The conditions for the effective hydrothermal transport of rare earth elements are the presence of anions (chloride, sulfate, carbonate) and alkalis [54–57]. It was previously established that potassic fluids preferentially transport heavy rare earth elements (HREE) relative to sodic fluids [55]. According to studies of Ivanyuk and co-authors [24], rischorrite was formed as a result of potassic metasomatism of foyaite under the influence of foidolites. It is likely that both during the formation of rischorrite by the metasomatic alteration of foyaite and during the formation of aluminous hornfelses by the metasomatic alteration of volcaniclastic and/or greenstone rocks, REEs were transported by potassic fluids in complexes with different ligands. (K,Na)-feldspar crystallization was the main reason for the decrease in the concentration of alkalis in the fluid. At the same time, rare earth elements were deposited in the form of highly insoluble monazite-(Ce) or xenotime-(Y).

The absence of metasomatic zoning within the xenoliths can be explained, firstly, by the relatively small size of the xenoliths and, secondly, by the nature of the influencing fluid. According to the experimental work of Preston and colleagues [58], melt compositions plotting towards nepheline on the nepheline–albite join are in equilibrium with fluids that are capable of converting a granite to a nepheline syenite at low fluid:rock ratios, whereas compositions towards the albite side of the join require very large fluid:rock ratios to perform this. In fact, small alkaline complexes of the ijolite–melteigite series have large metasomatic aureoles in relation to their size (e.g., Kovdor massif [59,60], Kola Peninsula, Russia) whereas nepheline syenite intrusions appear not to produce sizable aureoles (e.g., Khibiny massif). It should be noted that, despite the fact that foidolites are in contact with the host Archean gneisses in the Kovdor massif, extremely aluminous associations do not appear during the fenitization of gneisses. The reason, apparently, is the low content of fluorine in the foidolites of the Kovdor massif.

The mineral associations of aluminous hornfelses are evidence of the highly reducing conditions of their formation. Indeed, hercynite, sekaninaite, annite, fayalite, and ulvöspinel are rare minerals because they require unusually low oxygen fugacity ($fO_2$) to crystallize. Additional detailed studies are required to establish the reasons for the formation of the highly reduced mineral association of aluminous hornfelses. Here, we propose a possible mechanism based on the analogy of the process of serpentinization.

Serpentinization is a generalized term for retrograde metamorphism in ultramafic systems that produces one or more minerals in the serpentine group. An unusual feature of serpentinization is the production of fluids with remarkably low oxygen fugacity ($fO_2$) [61] and correspondingly high activities of reduced species such as $H_2$ and $CH_4$ [62]. Petrographic evidence for reducing conditions in serpentinites has been documented in the form of transition metal alloys and sulfides, common as accessory minerals in serpentinites [63]. Such alloys include awaruite ($Ni_3Fe$) and wairuite ($Co_3Fe$), both of

which contain substantial native iron, requiring very low $fO_2$ values approaching the iron–magnetite buffer [61]. Sulfides include heazlewoodite ($Ni_3S_2$), a reduced sulfide known only in serpentinites. Additionally, serpentinites have the lowest silica activity of common crustal rocks.

Frost and Beard [64] have shown that the distinctive petrological and geochemical properties of serpentinites (the highly magnetic nature, the reducing conditions, the calcic, high-pH fluids issuing from them) are all tied to the low silica activity of these rocks. First, low silica activity lowers the stability of the $Fe_3Si_2O_5(OH)_4$ component in serpentine, such that some of the ferrous iron in the primary olivine must go into magnetite. The formation of oxidized iron phases, especially magnetite, from the ferrous iron in the silicates is the root cause of the characteristically reduced conditions found in serpentinites.

Magnetite is a typical accessory mineral of aluminous hornfelses. It is possible that the critical deficiency of silica during the formation of some varieties of hornfelses was the cause of magnetite crystallization, which, in turn, was the cause of the reducing conditions.

## 6. Conclusions

1. Aluminous hornfelses, previously found in the central parts of the Khibiny and Lovozero massifs, were formed as a result of the influence on the protolith of fluids expelled from crystallizing foidolites and containing, in addition to alkalis, aluminum in the form of Na-Al-OH-F complexes. Thus, it is fluorine that controls the mobility of aluminum in the fluid and, consequently, the mineral associations of alkaline metasomatites.

2. The protolith of aluminous hornfelses cannot be unambiguously identified due to the high intensity of metasomatic alterations and the relatively small size of xenoliths. However, it can be argued that the protolith was extremely heterogeneous in terms of mineral and chemical composition.

3. The gain of alkalis and aluminum to rocks of protolith was the reason for the intense crystallization of (K,Na)-feldspar. As a result, a strong $SiO_2$ deficiency was formed, and Si-poor silicates (e.g., fayalite, $Al_2SiO_5$ polymorphs) and/or oxides (e.g., corundum, hercynite) crystallized.

**Supplementary Materials:** The following supporting information can be downloaded at: www.mdpi.com/article/10.3390/min12091076/s1, Table S1: Representative analyses of (K,Na)-feldspar; Table S2: Representative analyses of albite; Table S3: Representative analyses of the hercynite; Table S4: Representative analyses of the trioctahedral micas; Table S5: Representative analyses of the cordierite-sekaninaite; Table S6: Representative analyses of the fayalite; Table S7: Representative analyses of the garnets; Table S8: Representative analyses of the muscovite; Table S9: Major (wt. %) elements composition of the hornfelses; Table S10: REE (ppm) composition.

**Author Contributions:** J.A.M.: conceptualization, writing the manuscript and drawing figures; Y.A.P.: electron miscopy investigation, field works, revising the draft; N.G.K.: field works, providing samples, revising the draft; A.O.K.: discuss the results, revising the draft; V.N.Y.: field works, providing samples. All authors have read and agreed to the published version of the manuscript.

**Funding:** The field work was funded by the Ministry of Science and Higher Education of the Russian Federation, project no. AAAA-A19-119100290149-1. The microprobe, wet chemical, and ICP-MS analyses, and the X-ray diffraction (XRD) measurements were funded by Russian Science Foundation, project no. 21-47-09010.

**Data Availability Statement:** Not applicable.

**Acknowledgments:** We are grateful to the reviewers who helped us improve the presentation of our results.

**Conflicts of Interest:** The authors declare no conflicts of interest.

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
