# Peer review of "Fluorine Controls Mineral Assemblages of Alkaline Metasomatites"

_minerals, doi:10.3390/min12091076_

Round 1
Reviewer 1 Report
The paper “Fluorine controls mineral assemblages of alkaline metasomatites” by Mikhailova and co-authors is a significant contribution to the mineralogy and petrography of aluminous hornfels from xenoliths in the famous Khibiny and Lovozero massifs in Kola Peninsula, Russia.
The manuscript contains high-quality data. The authors studied several dozen of aluminous hornfelses in detail; more importantly, they determined their formation as a result of the metasomatic influence of foidolites. Moreover, they argued that the presence and high mobility of aluminum controlled by F in fluids derived from foidolites is reflected in the mineral associations of alkaline metasomatites.
This work is a good example of fieldwork and knowledge about the study area. The results are very interesting and are worth to be published in Minerals.
Overall, the language is quite good.
I have several minor comments, which are given below:
Line 26: Keywords – according to the Minerals guidelines, three to ten pertinent keywords can be added in this section. The authors proposed only three, but I think keywords like - aluminous hornfels, the main studied object, and also - Khibiny massif should be added. I do not know why it was omitted compared to Lovozero massif if most of the samples are from Khibiny massif.
Line 148: Description of Figure 2.
Replace: “Xenoliths of the aluminous hornfelses (1) among alkaline rocks (2) of the Khibiny and Lovozero massifs. (a) on the Kuivchorr (Lovozero massif); (b) on the Yudychvumchorr Mt. (Khibiny massif); the size of this xenolith is 1.5×0.6 meters; (c) on the Kaskasnyunchorr Mt. (Khibiny massif); the size of this xenolith is 100×50 meters. Photos by Gregory Ivanuyk” with “Xenoliths of the aluminous hornfelses (1) among alkaline rocks (2). (a) On the Kuivchorr (Lovozero massif); (b) on the Yudychvumchorr Mt. (Khibiny massif); the size of this xenolith is 1.5×0.6 meters; (c) on the Kaskasnyunchorr Mt. (Khibiny massif); the size of this xenolith is 100×50 meters. Photos by Gregory Ivanuyk”
I think it is unnecessary to mention twice the same massifs when you put the name of each of them in brackets on individual localities.
Lines 156-158: I have a problem with these two sentences. In the first one, the authors described quartzite and sandstone as two different but chemically similar rocks. I understood this sandstone in a bracket in the second sentence as a quartzite equivalent. What the authors mean by such a description???
Line 226: I think that (Ba) is missing after barite.
Line 252: Table 2
In accordance with the IMA list of minerals and work published by Warr, L.N. IMA-CNMNC Approved Mineral Symbols. Mineralogical Magazine 2021, 85, 291–320 authors given the mineral formulas and abbreviations. However, there are a few mistakes; the fluorite abbreviation is different and should be Flr, it also needs to be changed in some figures, and the formulas of albite, fayalite, and muscovite are incorrect.
Line 342: Maybe instead of “inclusions of grains or aggregates of grains of feldspar...” just simply replace it with “inclusion of feldspar...”
Line 383: Description of Figure 9 – replace “7d” with “9d”
Lines 387 and 391: “sekaninaite–cordierite series” and “cordierite-sekaninaite series” please unified.
Lines 427-429: “Andalusite and sillimanite are found in aluminous hornfelses both together in one sample and separately, but in the same xenolith, both associations with sillimanite and ones with andalusite can be present.”
This sentence is not understandable. What do you mean together and separately? You have a few samples from one xenolith, and in some you have only sillimanite, in others only andalusite, and in the next ones both? How does this relate to the crystallization conditions? Different parts of this xenolith have been formed in various P-T conditions? Please clarify it.
Lines 666 and 721: Replace “Khibina” with “Khibiny”
Remarks on the Figures:
The main comment on the graphical part is related to the abbreviations (Figures: 2, 4, 6, 7, 9). I do know why but the authors proposed some mineral abbreviations in red font with a black border. In several BSE images, for example, it is unreadable and unclear. I recommend traditional white font for dark spots and black for bright ones. Of course, it is only a suggestion, but in my opinion, the scale bar should be given deeper into the figure/photo/graph because in the present version, when it is on the verge of an image is not visible.
Remarks on reference list:
- Lines 802, 804, 805, 816(x4), 817, 820, 858(x2), 870: There was some program mistake or something else because, in several reference positions, there is a mistake related to the “surname” of many authors, mainly with the V letter. It is given in lowercase and should be capitalized.
- Line 815: Delete “2010”
- Lines 884 and 908: Give subscripts in oxide formulas and remove unnecessary spaces
- Please check the reference list again because I also might have missed something!
Author Response
We are very grateful for the high appreciation of our work and important comments. We took into account all the comments of the reviewer and made the appropriate changes in the text of the manuscript.
Point 1: Line 26: Keywords – according to the Minerals guidelines, three to ten pertinent keywords can be added in this section. The authors proposed only three, but I think keywords like - aluminous hornfels, the main studied object, and also - Khibiny massif should be added. I do not know why it was omitted compared to Lovozero massif if most of the samples are from Khibiny massif.
Response 1: Corrected. The following keywords have been added: Khibiny massif, aluminous hornfelses.
Point 2: Line 148: Description of Figure 2.
Replace: “Xenoliths of the aluminous hornfelses (1) among alkaline rocks (2) of the Khibiny and Lovozero massifs. (a) on the Kuivchorr (Lovozero massif); (b) on the Yudychvumchorr Mt. (Khibiny massif); the size of this xenolith is 1.5×0.6 meters; (c) on the Kaskasnyunchorr Mt. (Khibiny massif); the size of this xenolith is 100×50 meters. Photos by Gregory Ivanuyk” with “Xenoliths of the aluminous hornfelses (1) among alkaline rocks (2). (a) On the Kuivchorr (Lovozero massif); (b) on the Yudychvumchorr Mt. (Khibiny massif); the size of this xenolith is 1.5×0.6 meters; (c) on the Kaskasnyunchorr Mt. (Khibiny massif); the size of this xenolith is 100×50 meters. Photos by Gregory Ivanuyk”
Response 2: Corrected.
Point 3: Lines 156-158: I have a problem with these two sentences. In the first one, the authors described quartzite and sandstone as two different but chemically similar rocks. I understood this sandstone in a bracket in the second sentence as a quartzite equivalent. What the authors mean by such a description???
Response 3: Thank you for your comment. There was a mistake in the second sentence. Quartzite and sandstone are two different but chemically similar rocks. We have changed the second sentence as follows: "Tuffite with variable content of pyroclastic and terrigenous material is an intermediate rock between tuff on the one hand and quartzites and sandstones on the other hand".
Point 4: Line 226: I think that (Ba) is missing after barite.
Response 4: Corrected.
Point 5: Line 252: Table 2
In accordance with the IMA list of minerals and work published by Warr, L.N. IMA-CNMNC Approved Mineral Symbols. Mineralogical Magazine 2021, 85, 291–320 authors given the mineral formulas and abbreviations. However, there are a few mistakes; the fluorite abbreviation is different and should be Flr, it also needs to be changed in some figures, and the formulas of albite, fayalite, and muscovite are incorrect.
Response 5: Thank you for your comment. Corrected.
Point 6: Line 342: Maybe instead of “inclusions of grains or aggregates of grains of feldspar...” just simply replace it with “inclusion of feldspar...”
Response 6: Corrected.
Point 7: Line 383: Description of Figure 9 – replace “7d” with “9d”
Response 7: Replaсed.
Point 8: Lines 387 and 391: “sekaninaite–cordierite series” and “cordierite-sekaninaite series” please unified.
Response 8: Corrected.
Point 9: Lines 427-429: “Andalusite and sillimanite are found in aluminous hornfelses both together in one sample and separately, but in the same xenolith, both associations with sillimanite and ones with andalusite can be present.”
This sentence is not understandable. What do you mean together and separately? You have a few samples from one xenolith, and in some you have only sillimanite, in others only andalusite, and in the next ones both? How does this relate to the crystallization conditions? Different parts of this xenolith have been formed in various P-T conditions? Please clarify it.
Response 9: Thank you very much for your comment. In the same xenolith, samples were found containing both sillimanite in association with andalusite, and samples containing only sillimanite or only andalusite. Perhaps this is due to the fact that the P-T conditions for the formation of aluminous hornfelses were close to the sillimanite-andalusite equilibrium. We've changed the sentence and added a clarification.
Point 10: Lines 666 and 721: Replace “Khibina” with “Khibiny”
Response 10: Corrected.
Point 11: Remarks on the Figures:
The main comment on the graphical part is related to the abbreviations (Figures: 2, 4, 6, 7, 9). I do know why but the authors proposed some mineral abbreviations in red font with a black border. In several BSE images, for example, it is unreadable and unclear. I recommend traditional white font for dark spots and black for bright ones. Of course, it is only a suggestion, but in my opinion, the scale bar should be given deeper into the figure/photo/graph because in the present version, when it is on the verge of an image is not visible.
Response 11: The color of the abbreviation and the position of the scale bars in the figures 2, 4, 6, 7, 9 have been changed. Thank you for this comment. The figures look much better now.
Point 12: Remarks on reference list:
- Lines 802, 804, 805, 816(x4), 817, 820, 858(x2), 870: There was some program mistake or something else because, in several reference positions, there is a mistake related to the “surname” of many authors, mainly with the V letter. It is given in lowercase and should be capitalized.
- Line 815: Delete “2010”
- Lines 884 and 908: Give subscripts in oxide formulas and remove unnecessary spaces
Please check the reference list again because I also might have missed something!
Response 12: References have been checked and corrected.

Reviewer 2 Report
line 72: Not sure that "assumptions" is the correct word here
line 108: What do you mean by "day surface"?
line 138: Where is the minor ring in Fig 1b? I can't see it
line 226: Accepted IMA names are celestine and baryte
line 253: Is this the Warr list? If yes, please cite Warr
line 278: No need for the « » symbols
line 293: This is an important outcrop - can you add the coordinates of this place?
line 301: "hornfels minerals"?
line 306: Add a comma before "and ubiquitous minerals"
line 319: "..and feldspar" - which feldspar? K-feldspar?
line 324: plagioclase? if An goes to 48...
line 352, 391: analyses
line 354, 371: Don't use red-green colour combinations. It is the most common type of colour blindness. Use different colours (orange and purple work well), and also use symbol (circle, triangle) to distinguish the massifs.
line 378: You use "Po" for pyrrhotite I presume? You used a different abbreviation in the previous figure. Be consistent please.
line 628: "...besides the gain..."
line 652: etc
line 665, 667: What is Na2+?
line 666: Khibina or Khibiny?
line 702: Not accurate. They are only effective in highly alkaline solutions, see for example https://doi.org/10.1126/sciadv.abb6570 and https://doi.org/10.1038/s41467-022-28943-z and read the papers you cite carefully, including their newer work https://doi.org/10.1016/j.chemgeo.2016.06.005 saying that F does not mobilise REE
line 734: I don't understand this paragraph. Which way does the reaction go? Where is native iron in these schemes? What does oxygen do in your reactions? Is it a reactant, product, or a proxy for oxygen fugacity? What controls, or the driving force for these reactions? Your reactions include aegirine, but you did not report aegirine in your rocks. This is a crucial part of your work, and I think it deserves much more treatment than just one paragraph (which honestly I did not understand). How is silica deficiency demonstrated by these reactions?
Author Response
We took into account all the comments of the reviewer and made the appropriate changes in the text of the manuscript.
Point 1: line 72: Not sure that "assumptions" is the correct word here
Response 1: The word "assumptions" was replaced by the word "conclusions".
Point 2: line 108: What do you mean by "day surface"?
Response 2: "Day surface" means the top of the massif. This sentences has been modified as follows: "The laccolith has a size 20×30 km at the top, and about 12×16 km on a 5 km depth".
Point 3: line 138: Where is the minor ring in Fig 1b? I can't see it
Response 3: In Figure 1b, the designations "Minor Ring" and "Main foidolite Ring" are added.
Point 4: line 226: Accepted IMA names are celestine and baryte
Response 4: Corrected.
Point 5: line 253: Is this the Warr list? If yes, please cite Warr
Response 5: An article (Warr, L.N. IMA-CNMNC Approved Mineral Symbols. Mineralogical Magazine 2021, 85, 291–320, doi:10.1180/mgm.2021.43) has been added to the references.
Point 6: line 278: No need for the « » symbols
Response 6: Symbols « » removed.
Point 7: line 293: This is an important outcrop - can you add the coordinates of this place?
Response 7: The coordinates of points Е-97-45 (67039'56.5''N 33058'35.4''E) and Е-97-1 (67039'59.6''N 33058'56.5''E) are added to the Figure 5 caption.
Point 8: line 301: "hornfels minerals"?
Response 8: Yes. "Hornfelses minerals" replaced with "hornfels minerals".
Point 9: line 306: Add a comma before "and ubiquitous minerals"
Response 9: Added.
Point 10: line 319: "..and feldspar" - which feldspar? K-feldspar?
Response 10: (K,Na)-feldspar. This sentence has been modified as follows: "Such perthites consist of pure albite (Ab99-100) and (K,Na)-feldspar which the sodium content is lower than in unaltered (K,Na)-feldspar."
Point 11: line 324: plagioclase? if An goes to 48...
Response 11: The word "albite" was changed to "plagioclase"
Point 12: line 352, 391: analyses
Response 12: Corrected.
Point 13: line 354, 371: Don't use red-green colour combinations. It is the most common type of colour blindness. Use different colours (orange and purple work well), and also use symbol (circle, triangle) to distinguish the massifs.
Response 13: In figures 6c and 8a-d, the color and shape of the dots are changed.
Point 14: line 378: You use "Po" for pyrrhotite I presume? You used a different abbreviation in the previous figure. Be consistent please.
Response 14: Corrected.
Point 15: line 628: "...besides the gain..."
Response 15: Corrected.
Point 16: line 652: etc
Response 16: Corrected.
Point 17: line 665, 667: What is Na2+?
Response 17: This is mistake. Na2+ corrected to Na+.
Point 18: line 666: Khibina or Khibiny?
Response 18: Khibiny is the correct spelling. Errors in the text have been corrected.
Point 19: line 702: Not accurate. They are only effective in highly alkaline solutions, see for example https://doi.org/10.1126/sciadv.abb6570 and https://doi.org/10.1038/s41467-022-28943-z and read the papers you cite carefully, including their newer work https://doi.org/10.1016/j.chemgeo.2016.06.005 saying that F does not mobilise REE
Response 19: Thank you very much for this comment. I carefully read both the cited literature and the articles recommended by the reviewer. Indeed, there was a serious mistake in the text of the article. The section of the discussion concerning the behavior of rare earth elements has been completely rewritten (lines 703-715).
Point 20: line 734: I don't understand this paragraph. Which way does the reaction go? Where is native iron in these schemes? What does oxygen do in your reactions? Is it a reactant, product, or a proxy for oxygen fugacity? What controls, or the driving force for these reactions? Your reactions include aegirine, but you did not report aegirine in your rocks. This is a crucial part of your work, and I think it deserves much more treatment than just one paragraph (which honestly I did not understand). How is silica deficiency demonstrated by these reactions?
Response 20: Thanks for the important comment. I fully agree that this section was written incomprehensibly. I have now completely rewritten this part of the discussion (lines 732-755).
